# Emergence of trait variability through the lens of nitrogen assimilation in *Prochlorococcus*

**Paul M Berube[1]\*, Anna Rasmussen[1†], Rogier Braakman[1], Ramunas Stepanauskas[2], Sallie W Chisholm[1,3]**

[1]Department of Civil and Environmental Engineering, Massachusetts Institute of Technology, Cambridge, United States; [2]Bigelow Laboratory for Ocean Sciences, East Boothbay, United States; [3]Department of Biology, Massachusetts Institute of Technology, Cambridge, United States

**Abstract** Intraspecific trait variability has important consequences for the function and stability of marine ecosystems. Here we examine variation in the ability to use nitrate across hundreds of *Prochlorococcus* genomes to better understand the modes of evolution influencing intraspecific allocation of ecologically important functions. Nitrate assimilation genes are absent in basal lineages but occur at an intermediate frequency that is randomly distributed within recently emerged clades. The distribution of nitrate assimilation genes within clades appears largely governed by vertical inheritance, gene loss, and homologous recombination. By mapping this process onto a model of *Prochlorococcus'* macroevolution, we propose that niche-constructing adaptive radiations and subsequent niche partitioning set the stage for loss of nitrate assimilation genes from basal lineages as they specialized to lower light levels. Retention of these genes in recently emerged lineages has likely been facilitated by selection as they sequentially partitioned into niches where nitrate assimilation conferred a fitness benefit.
DOI: https://doi.org/10.7554/eLife.41043.001

\*For correspondence:
pmberube@gmail.com

Present address: †Department of Earth System Science, Stanford University, Palo Alto, United States

Competing interests: The authors declare that no competing interests exist.

## Introduction

*Prochlorococcus* and its closest relative, *Synechococcus*, are non-nitrogen-fixing cyanobacteria whose common ancestor emerged approximately 823–644 Mya (*Sánchez-Baracaldo et al., 2014*). They are among the most abundant photosynthetic organisms on the planet and are estimated to be jointly responsible for 25% of ocean net primary productivity (*Flombaum et al., 2013*). While *Synechococcus* has a broader geographic range, *Prochlorococcus* is primarily restricted to the tropical and subtropical open ocean where nutrients are scarce. Genomic signatures of adaptation to these highly oligotrophic environments are found across *Prochlorococcus* genomes (*Rocap et al., 2003*; *Martiny et al., 2006*; *Kettler et al., 2007*) and reveal the selection pressures operating on populations in the wild (*Malmstrom et al., 2010*; *Coleman and Chisholm, 2010*; *Rusch et al., 2010*).

Access to nitrogen – often the proximal limiting nutrient for phytoplankton growth across the global ocean (*Tyrrell, 1999*) – is a nearly ubiquitous challenge facing *Prochlorococcus*. All *Prochlorococcus* can assimilate ammonium, but the remainder of nitrogen uptake pathways – for amino acids, urea, cyanate, nitrite, and nitrate – are encoded by flexible genes that are found in some, but not all, *Prochlorococcus* (*Biller et al., 2015*). Nitrate assimilation is of particular interest because nitrate is abundant at the base of the sunlit euphotic zone and can fuel productivity in nitrogen limited surface waters during the vertical advection of this deeper water (*Dugdale and Goering, 1967*; *Eppley et al., 1979*; *Johnson et al., 2010*). Nearly all observed marine *Synechococcus* have the genes encoding the transporters, reductases, and molybdopterin cofactor biosynthesis proteins

required for nitrate assimilation (*Scanlan et al., 2009*). Most *Prochlorococcus*, however, lack this genetic repertoire (*Moore et al., 2002*; *García-Fernández et al., 2004*; *Berube et al., 2015*). Given that the nitrogen atom of nitrate is fully oxidized, nitrate is an expensive source of nitrogen for the cell. Compared to highly reduced nitrogen sources such as ammonium and urea, it has been well documented that cells must compensate for the increase in photochemically generated reducing power required for nitrate assimilation (*Kok, 1952*; *van Oorschot, 1955*; *Myers, 1980*; *Falkowski and Stone, 1975*; *Raven, 1988*; *Thompson et al., 1989*). In *Prochlorococcus*, these costs appear to be observed as a decrease in growth rate under saturating light intensity when supplied with nitrate compared to ammonium as the sole nitrogen source (*Berube et al., 2015*).

*Prochlorococcus* has diverged into multiple clades, many with adaptations that can be mapped onto gradients of key environmental variables (*Biller et al., 2015*). Among the dozens of cultured *Prochlorococcus* strains sequenced from these clades, genes encoding nitrate assimilation proteins have only been observed in a few strains belonging to the high-light adapted HLII clade and the low-light adapted LLI clade (*Berube et al., 2015*). In the open ocean, the distribution of *Prochlorococcus* with the potential to assimilate nitrate shows distinct seasonal patterns, reaching a peak in abundance in the summer when the solar energy supply is greater and overall nitrogen concentrations are lower (*Berube et al., 2016*).

Within the HLII clade – often the most abundant *Prochlorococcus* clade in subtropical gyres by an order of magnitude (*Malmstrom et al., 2010*) – the frequency of cells that are capable of assimilating nitrate (up to 20–50%) is positively correlated with decreased nitrogen availability in surface waters where they dominate. This has suggested that cells with access to a wide pool of nitrogen sources and a ready supply of energy are likely at a selective advantage when nitrogen is limiting (*Berube et al., 2016*). For the LLI clade, the trade-offs appear more complex. All previously described *Prochlorococcus* in the LLI clade can assimilate nitrite, but only a fraction can also assimilate the more oxidized nitrate (*Berube et al., 2015*). These cells are found in the mid-euphotic zone and near the nitracline where inorganic nitrogen concentrations begin to rise with increasing depth. The depth distribution of these low-light adapted cells is associated with a peak in nitrite concentration (*Berube et al., 2016*). In stratified marine systems, the depth of this nitrite maximum layer is commonly correlated with the nitracline and the depth at which photosynthetically active radiation is attenuated to ~1% of surface irradiance (*Lomas and Lipschultz, 2006*). At this inflection point in the water column, cells have access to inorganic nitrogen sources of varying redox state ($NO_3^-$, $NO_2^-$, and $NH_4^+$), but must also compete with ammonia and nitrite oxidizing microorganisms for access to reduced and thus more easily assimilated nitrogen sources (*Berube et al., 2016*). Frequency-dependent selection processes (*Cordero and Polz, 2014*) may have a role in selecting for the optimal distribution of nitrogen assimilation traits across these low-light adapted *Prochlorococcus* populations.

While we have gained general insights into the tradeoffs governing the overall abundance of *Prochlorococcus* cells capable of nitrate assimilation in the wild, genomic data have thus far painted an incomplete picture of the evolution of this trait at the genomic and sequence levels. Although the gene order and genomic location of the nitrate assimilation gene cluster among HLII clade genomes is highly conserved (*Martiny et al., 2009*; *Berube et al., 2015*), there is evidence that one genome acquired duplicate copies of these genes via phage-mediated gene transfer (*Berube et al., 2015*). Comparative genomics of a small number of cultured strains suggested that *Prochlorococcus* lost the nitrate assimilation genes following their divergence from *Synechococcus* and then regained them early in the emergence of the LLI and HLII clades (*Berube et al., 2015*). Examination of GC content, gene synteny, and local genomic architecture, however, suggested that nitrate assimilation genes may have descended vertically in *Prochlorococcus* (*Berube et al., 2015*). Overall, the relative roles of vertical descent, gene duplication, genome streamlining, and horizontal gene transfer in mediating the evolution of these genes is thus ambiguous – especially given the limited number of cultivated representatives among many *Prochlorococcus* clades.

In this study, we examined a genomics data set consisting of 486 *Prochlorococcus* and 59 *Synechococcus* genomes from both cultivated isolates and wild single cells to better constrain the relative impacts of different evolutionary forces on the diversification of nitrate assimilation genes in *Prochlorococcus*. Drawing on our understanding of the physiology and ecology of major phylogenetic groups of *Prochlorococcus*, we also explore how evolution gives rise to a heterogeneous distribution of the nitrate assimilation trait across the *Prochlorococcus* genus.

## Results and discussion

### Nitrate assimilation genes are found within a distinct subset of *Prochlorococcus* clades

To constrain the higher order relationship between nitrate assimilation and major clades of *Prochlorococcus*, we first assessed the distribution of genes involved in nitrate and nitrite assimilation across a core marker gene phylogeny for *Prochlorococcus* and *Synechococcus* that includes both isolate and single cell genomes (*Figure 1*). Given the inherent incompleteness of single cell genome assemblies, we also developed a PCR assay to screen amplified single cell DNA for the presence of *narB* (nitrate reductase), a key gene distinguishing the complete pathway for nitrate assimilation from the downstream half for nitrite assimilation alone. We found that while genes for nitrite assimilation are distributed across the tree of *Prochlorococcus*, thus enabling the assimilation of the more reduced nitrite, genes for the full pathway of nitrate assimilation are restricted to particular clades (*Figure 1*). Among the high-light adapted *Prochlorococcus*, *narB* was found in the HLI, HLII, and HLVI clades (*Figure 1*). It also appears rare for high-light adapted *Prochlorococcus* genomes to encode only the downstream half of the nitrate assimilation pathway – the reduction of nitrite to ammonium – instead of the complete pathway (*Figure 1*). Among the polyphyletic group of low-light adapted *Prochlorococcus*, *narB* was exclusively found in the LLI clade (*Figure 1*). We therefore conclude that the complete pathway for nitrate assimilation is essentially restricted to clades that emerged more recently in the evolution of *Prochlorococcus*.

The apparent absence of *narB* in some *Prochlorococcus* clades is consistent with expectations based on tradeoffs (e.g. energetic and trace metal requirements) that are inferred to govern nitrate assimilation. The HLIII and HLIV clades, which lack this trait in available single cell genome assemblies (*Malmstrom et al., 2013*), are generally characterized by adaptations to iron limitation (*Rusch et al., 2010*; *Malmstrom et al., 2013*) and may thus be under selective pressure to dispense with the nitrate assimilation pathway due to its high iron requirement (*Raven, 1988*). Among all isolate and single cell genome assemblies for low-light adapted *Prochlorococcus* in our data set (82% average genome recovery), 73% contained the genes for the downstream half of the nitrate assimilation pathway – encoding machinery for the transport and reduction of nitrite to ammonium (*Figure 1*). We note that this fraction represents a lower bound estimate given the inherent incompleteness of single cell genome assemblies. While *nirA* (nitrite reductase) was broadly distributed across low-light adapted *Prochlorococcus* genomes, only those belonging to the LLI clade were observed to also possess *narB* (*Figure 1*). Cells belonging to this clade dominate at shallower depths than other low-light adapted cells and moreover expand into the surface mixed layer during winter months (*Zinser et al., 2007*; *Malmstrom et al., 2010*). Thus, among low-light adapted *Prochlorococcus*, the LLI clade experiences higher irradiance levels which would facilitate the generation of required reducing power equivalents to support the reduction of nitrate. We further observed that all sequenced isolates belonging to the LLI clade possess *nirA* and the average genome recovery of isolate and single cell genome assemblies belonging to the LLI clade matched the proportion of these assemblies with an annotated *nirA* gene (81% average genome recovery; 81% containing *nirA*). These data suggest that the ability to assimilate the more reduced nitrite is a core trait for cells belonging to the LLI clade. We suspect that this may be a selective advantage for *Prochlorococcus* that live in close proximity to elevated concentrations of nitrite (*Berube et al., 2016*) that occur in the ubiquitous primary nitrite maximum (*Lomas and Lipschultz, 2006*).

### Nitrate assimilation genes do not co-vary with other flexible genes

Identifying traits that are under- or over-represented in *Prochlorococcus* genomes that harbor nitrate assimilation genes could shed additional light on other selection pressures similarly operating on cells with this trait and how cells in these clades balance the tradeoffs of the nitrate assimilation pathway. To explore this, we compared the flexible gene content of single cell *Prochlorococcus* genomes that possess *narB* with those that lack *narB* using gene enrichment analysis. No flexible genes other than those involved in nitrate assimilation were found to be over- or under-represented in genomes containing *narB* (hypergeometric test with Benjamini and Hochberg correction; p<0.05) (*Figure 2*, *Figure 2—figure supplement 1*). Although there appears to be a selective advantage to carrying nitrate assimilation genes under nitrogen limiting conditions (*Berube et al., 2016*), cells

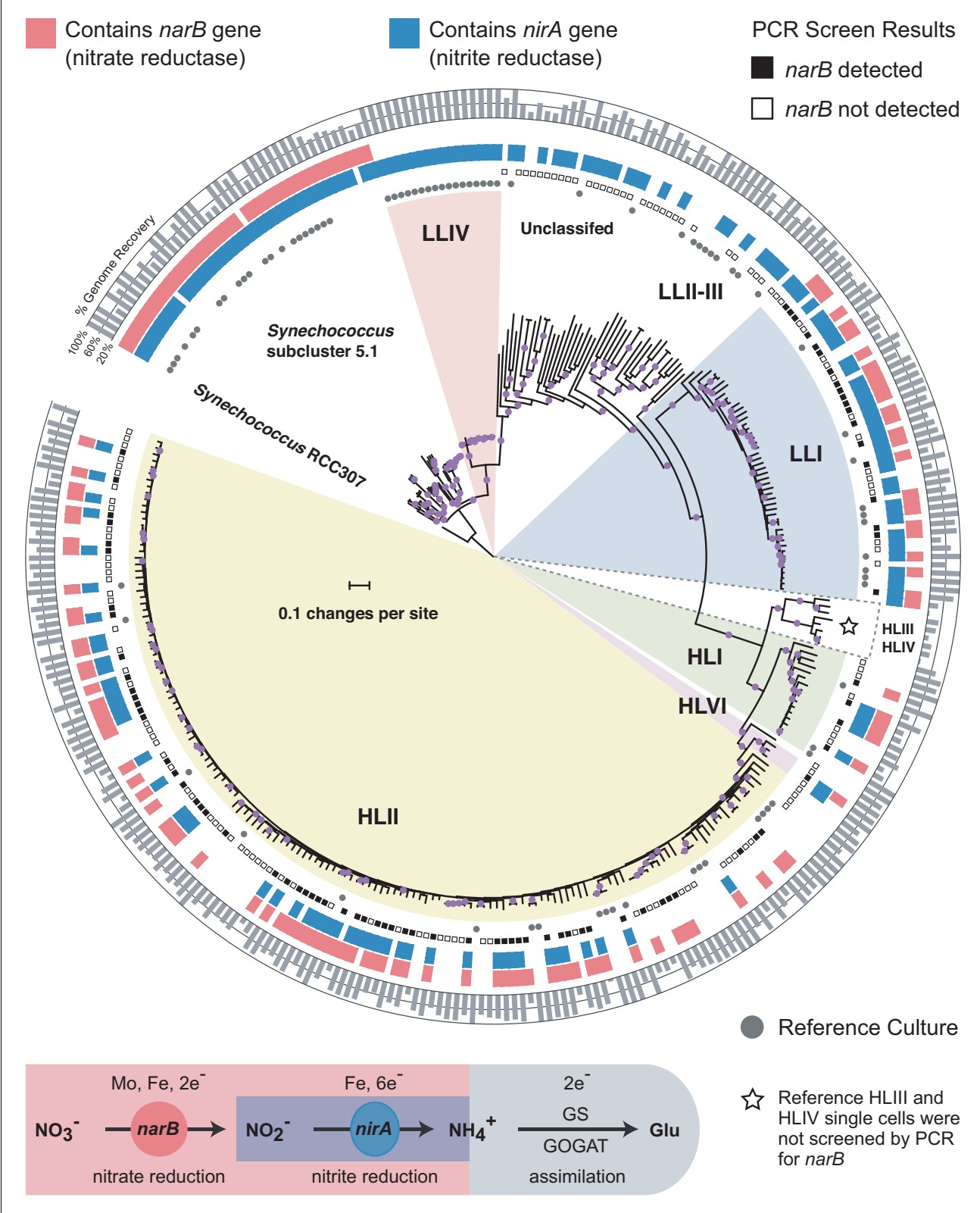

**Figure 1.** Distributions of the nitrate reductase (*narB*) and nitrite reductase (*nirA*) genes across a core marker gene phylogeny of 329 *Prochlorococcus* and *Synechococcus* genomes. Selected *Prochlorococcus* clades are highlighted as pie slices. Red bars indicate genomes with the potential to use nitrate based on the presence/absence of a *narB* gene. Blue bars indicate genomes with an annotated *nirA* gene in the genome assembly. The outer ring indicates the estimated percentage of the genomes recovered as a gray bar chart. Reference culture genomes are indicated by gray circles and

*Figure 1 continued on next page*

*Figure 1 continued*

the results of the PCR screen for *narB* are indicated by filled (present) and open (absent) squares. Single cells belonging to the HLIII and HLIV clades of *Prochlorococcus* were not screened by PCR and are only included as additional reference genomes. The nucleotide phylogeny is based on a concatenated alignment of 37 marker genes in the PhyloSift software package and inferred using maximum likelihood in RAxML (GTRCAT model) with automatic bootstopping criteria (250 replicate trees). Filled purple circles on branches indicate that the associated genomes clustered together in at least 75% of trees.

DOI: https://doi.org/10.7554/eLife.41043.002

The following source data is available for figure 1:

**Source data 1.** Compressed tar archive (zip format) containing the concatenated codon alignment (fasta format) and tree file (newick format) used to generate *Figure 1*.

DOI: https://doi.org/10.7554/eLife.41043.003

capable of using nitrate are no more likely to carry additional accessory nitrogen assimilation pathways such as those for cyanate, urea, or amino acid assimilation. Overall, it appears that the nitrate assimilation genes have evolved independently of other flexible traits in *Prochlorococcus*.

## Nitrate assimilation genes are rarely acquired by *Prochlorococcus* through non-homologous recombination mechanisms

Trait absence in basal lineages but presence in recently emerged lineages suggests a possible role for horizontal gene transfer: *Prochlorococcus* may have lost the nitrate assimilation genes early after its divergence from *Synechococcus* and reacquired them later through horizontal gene transfer mechanisms (*Berube et al., 2015*). Other evidence, however, namely the close relationship between gene and whole genome GC contents as well as conservation in the location and gene order of the nitrate assimilation gene cluster, suggests that vertical descent has been an important factor in shaping these genome features (*Martiny et al., 2009*; *Berube et al., 2015*). To further assess the role of horizontal acquisition in the evolution of the nitrate assimilation trait in *Prochlorococcus*, we first examined in greater depth the patterns of diversity of this trait within clades.

We compared the phylogenies of individual proteins in both the upstream and downstream halves of the nitrate assimilation pathway with those of core marker proteins (primarily ribosomal proteins) that we assume are vertically inherited. Phylogenies for the nitrate reductase (NarB) and a molybdopterin biosynthesis protein (MoaA) were generally congruent with the core protein phylogeny at the clade level, arguing against frequent horizontal gene transfer between clades (*Figure 3*, panels a,c,d). The nitrate transporter (NapA) protein phylogeny (*Figure 3*, panel b) exhibited a branching pattern that grouped LLI *Prochlorococcus* with *Synechococcus*, but the corresponding gene phylogeny of *napA* was, however, congruent with the core gene phylogeny (*Figure 3—figure supplement 1*, panel b). Regardless, both HL and LLI *Prochlorococcus* strains possessed proteins in the upstream half of the nitrate assimilation pathway that were monophyletic and we conclude that recent gene transfer between clades has been minimal or absent.

Proteins involved in the downstream half of the pathway – responsible for the transport and reduction of nitrite and encoded by a functional cassette with three genes (*focA*, *nirA*, and *nirX*) – had phylogenies that were also largely consistent with the core protein phylogeny (*Figure 3*, panels a,e,f). But, we noticed one exception: The high-light adapted AG-363-P06 single cell (HLVI clade) possesses genes encoding a nitrite transporter (FocA) and a nitrite reductase (NirA) that are similar to low-light adapted *Prochlorococcus* (*Figure 3*, panels e,f). Among the single cells in our data set, this is the only genome belonging to a high-light adapted clade that contains the *focA* gene. This suggests that this HLVI *Prochlorococcus* strain recently acquired the nitrite assimilation cassette from a low-light adapted *Prochlorococcus*.

We next examined closed genomes and individual contigs from single cells to look for aberrations in the location and gene order of the nitrate assimilation gene cluster that might suggest gene acquisition through non-homologous recombination mechanisms. Within the HLII clade, there were 28 genome assemblies containing the complete nitrate assimilation gene cluster with sufficient sequence data extending into adjacent genomic regions on a single contig. In 27 of them, the nitrate assimilation genes were found in a core syntenic genomic region between a conserved gene encoding a sodium-dependent symporter (CyCOG_60001297) and *polA*, encoding DNA polymerase I

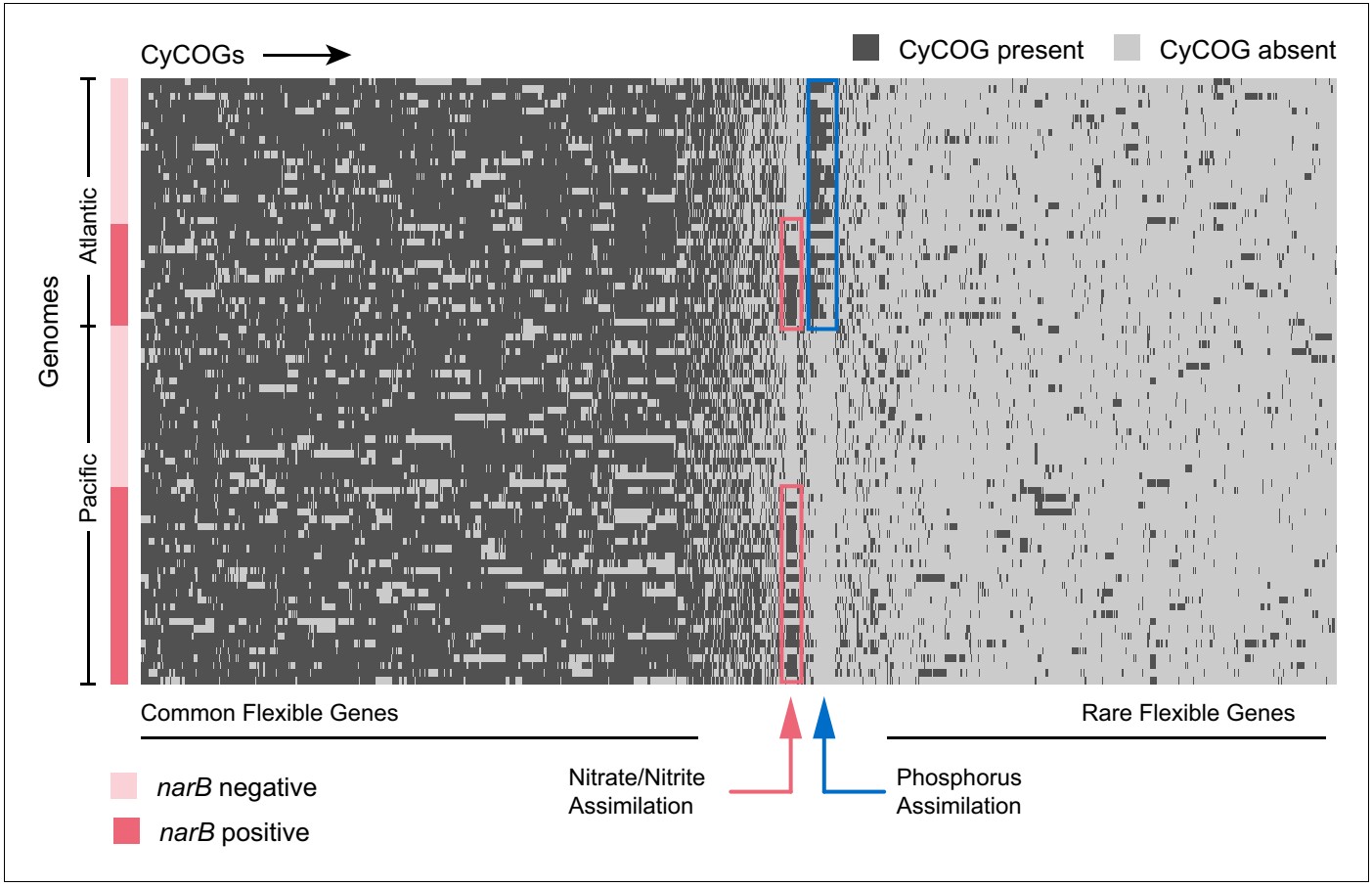

**Figure 2.** Hierarchical clustering of presence and absence distributions for flexible CyCOGs found in 83 *Prochlorococcus* HLII single cell genomes with genome recoveries of at least 75% (median 90%). Genomes are sorted by Atlantic and Pacific Oceans and by the presence/absence of the *narB* gene – a marker for the capacity to assimilate nitrate. Other than genes in the nitrate assimilation gene cluster (green box), no CyCOGs were over- or under-represented among the flexible genes in genomes containing narB. Genomes from the Atlantic, regardless of whether or not they contained the *narB* marker gene, were enriched in phosphorus assimilation genes (blue box).

DOI: https://doi.org/10.7554/eLife.41043.004

The following source data and figure supplement are available for figure 2:

**Source data 1.** Binary matrix containing the raw presence and absence data for each CyCOG in each genome analyzed for *Figure 2* and *Figure 2—figure supplement 1*.

DOI: https://doi.org/10.7554/eLife.41043.006

**Source data 2.** Compressed tar archive (zip format) containing input and output files for gene enrichment analysis using BiNGO 3.0.3 (*Maere et al., 2005*) in Cytoscape 3.4 (*Shannon et al., 2003*).

DOI: https://doi.org/10.7554/eLife.41043.007

**Figure supplement 1.** Hierarchical clustering of presence and absence distributions for flexible CyCOGs found in 22 *Prochlorococcus* single cell genomes belonging to the LLI clade with genome recoveries of at least 75% (median 87%).

DOI: https://doi.org/10.7554/eLife.41043.005

(*Figure 4*, *Figure 4—figure supplement 1*). The remaining HLII genome, MIT0604, is the only documented member of the HLII clade with duplicate copies of the nitrate assimilation gene cluster located in separate genomic islands (*Berube et al., 2015*).

In two single cell genomes belonging to the HLI and HLVI clades, the complete nitrate assimilation gene cluster is found in a different genomic region than in cells belonging to the HLII clade (*Figure 4*, *Figure 4—figure supplement 2*). Thus, it appears that the genomic position of these genes is largely conserved within individual high-light adapted clades but can differ between high-light adapted clades. Regardless of their genomic location, gene order was conserved in all genomes belonging to this monophyletic group of high-light adapted clades (*Figure 4*). The cell belonging to

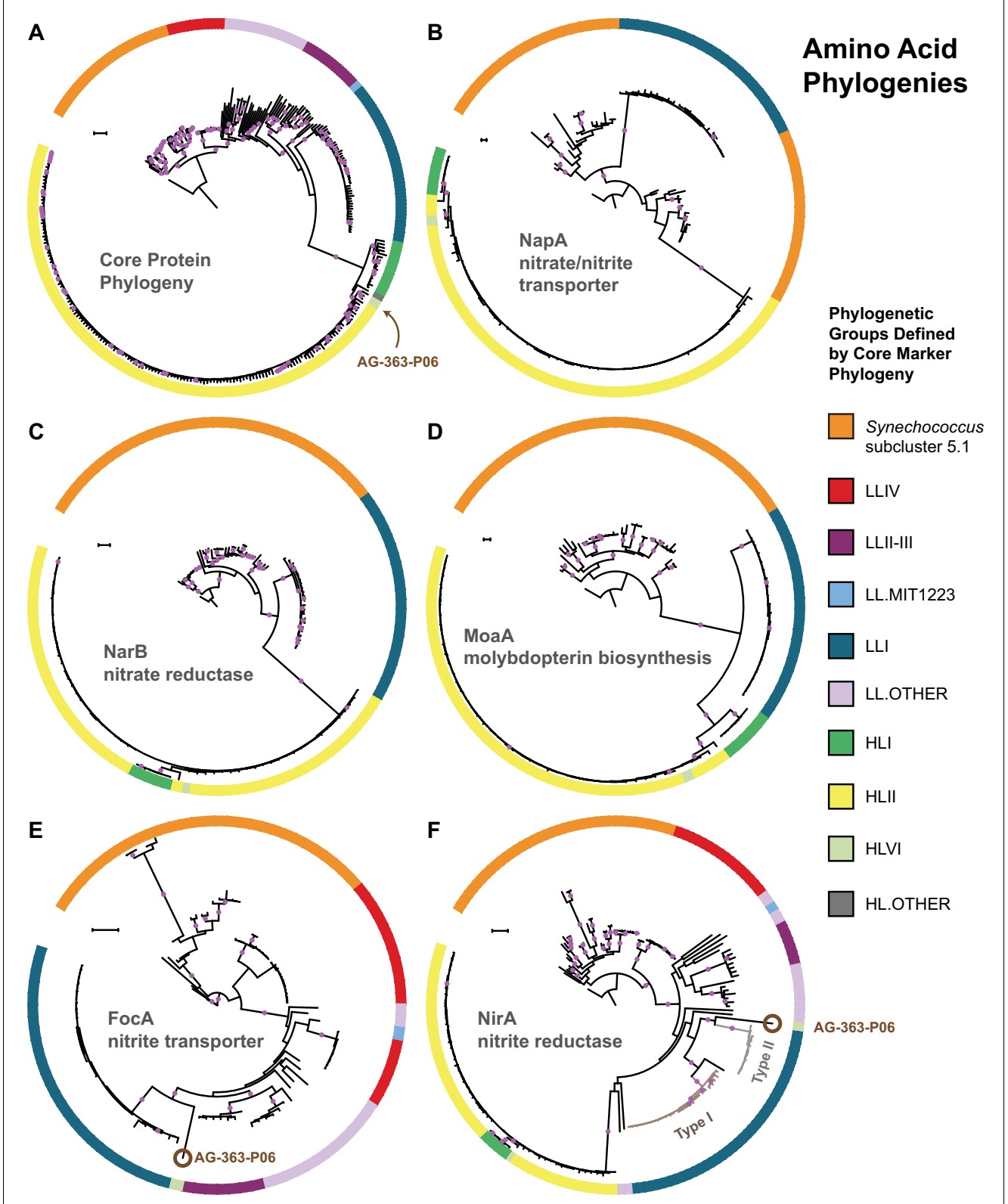

**Figure 3.** The core marker protein phylogeny of *Prochlorococcus* and *Synechococcus*. (A) in comparison to phylogenies for the nitrate/nitrite transporter, NapA (B), the nitrate reductase, NarB (C), the molybdopterin biosynthesis protein, MoaA (D), the nitrite transporter, FocA (E), and the nitrite reductase, NirA (F). Interclade horizontal gene transfer is minimal for genes encoding proteins in the upstream half of the nitrate assimilation pathway (B–D) since clades defined by the core phylogeny (A) remain separate. Horizontal gene transfer is observed in a few instances for genes

*Figure 3 continued on next page*

*Figure 3 continued*

encoding proteins in the downstream half of the nitrate assimilation pathway (**E–F**). One single cell (AG-363-P06; brown circle) from the high-light adapted HLVI clade possesses FocA and NirA proteins most similar to those from low-light adapted *Prochlorococcus*. *Prochlorococcus* belonging to the LLI clade possess two types of NirA as indicated by well supported phylogenetic divergence of NirA among LLI cells. Filled purple circles on branches indicate that the associated taxa clustered together in at least 75% of trees. Scale bars are 0.1 changes per site.
DOI: https://doi.org/10.7554/eLife.41043.008
The following source data and figure supplement are available for figure 3:

**Source data 1.** Compressed tar archive (zip format) containing codon alignments (fasta format) and tree files (newick format) used to generate *Figure 3* and *Figure 3—figure supplement 1*.
DOI: https://doi.org/10.7554/eLife.41043.010
**Figure supplement 1.** The core marker gene phylogeny of *Prochlorococcus* and *Synechococcus*.
DOI: https://doi.org/10.7554/eLife.41043.009

the HLVI clade (AG-363-P06) that only has the downstream half of the pathway – which we argue above was acquired from low-light adapted *Prochlorococcus* (*Figure 3*) – encodes these genes in an entirely different region located upstream of the ribosomal RNA operon (*Figure 4*, *Figure 4—figure supplement 2*). In this one case, it is likely that gene acquisition was driven by non-homologous recombination mechanisms.

Among 28 LLI clade genomes, we found that the nitrate and nitrite assimilation genes were always located in a core syntenic region between the pyrimidine biosynthesis gene, *pyrG*, and the polyphosphate kinase gene, *ppk* (*Figure 4*, *Figure 4—figure supplement 3*), which corresponds with the position of these genes in *Synechococcus* (*Rocap et al., 2003*; *Martiny et al., 2009*; *Berube et al., 2015*), strengthening an argument for vertical inheritance of these genes during the evolution of low-light adapted *Prochlorococcus*. Regardless of whether the LLI clade genomes encode the complete nitrate assimilation pathway or only the downstream half of the pathway (*Figure 4*), the order of the genes is conserved (*Figure 4*) with one exception: In the AG-311-K21 single cell there was an apparent deletion of the transporters and molybdopterin biosynthesis genes, leaving only the *nirA* and *narB* reductase genes (*Figure 4*). Given that both the *pyrG* and *ppk* genes were present in their expected locations in this genome, this is probably a genuine deletion and may reflect initial stages in the loss of the nitrate assimilation pathway in this LLI clade cell. Note, however, that this single cell genome assembly is 46% complete and thus we cannot rule out rearrangement of the missing genes to unassembled regions of the genome. Finally, there is a general pattern in which a small number of LLI genomes lack the nitrite transporter gene, *focA*, and possess a distinct version (Type II) of the nitrite reductase gene, *nirA* (*Figure 3*, *Figure 3—figure supplement 1*, *Figure 4*). These divergent enzymes could be a result of fine-scale niche partitioning, and biochemical studies are warranted to understand their potential functional and ecological significance.

## Homologous recombination shapes the underlying diversity of nitrate assimilation genes in *Prochlorococcus*

Thus far, our analysis has provided little evidence for acquisition of nitrate assimilation genes by *Prochlorococcus* through non-homologous recombination-based mechanisms. Homologous recombination between closely related cells, however, can facilitate the gain and loss of genes through recombination in core genomic regions that flank the genes in question (*Apagyi et al., 2018*; *Oliveira et al., 2017*) and can further act to limit the genetic divergence of shared loci (*Andam et al., 2010*; *Andam and Gogarten, 2011*; *Rosen et al., 2015*). To explore the potential role of this process, we evaluated the relative influence of recombination and mutation on shaping the diversity of genomic regions containing the nitrate assimilation gene cluster. We found that r/m values (the ratio of nucleotide changes due to recombination relative to point mutation) were well in excess of 1 for both the nitrate assimilation genes and the core genomic regions that flank them (*Table 1*), indicating that homologous recombination has a role in modulating diversity within these genomic regions. It is being increasingly acknowledged that bacteria can behave like sexually recombining populations (*Fraser et al., 2007*) and that recombination has a role in maintaining overall population diversity and in facilitating gene-specific, rather than genome-wide, sweeps within populations (*Shapiro et al., 2012*; *Shapiro, 2016*; *Rosen et al., 2015*). High rates of homologous

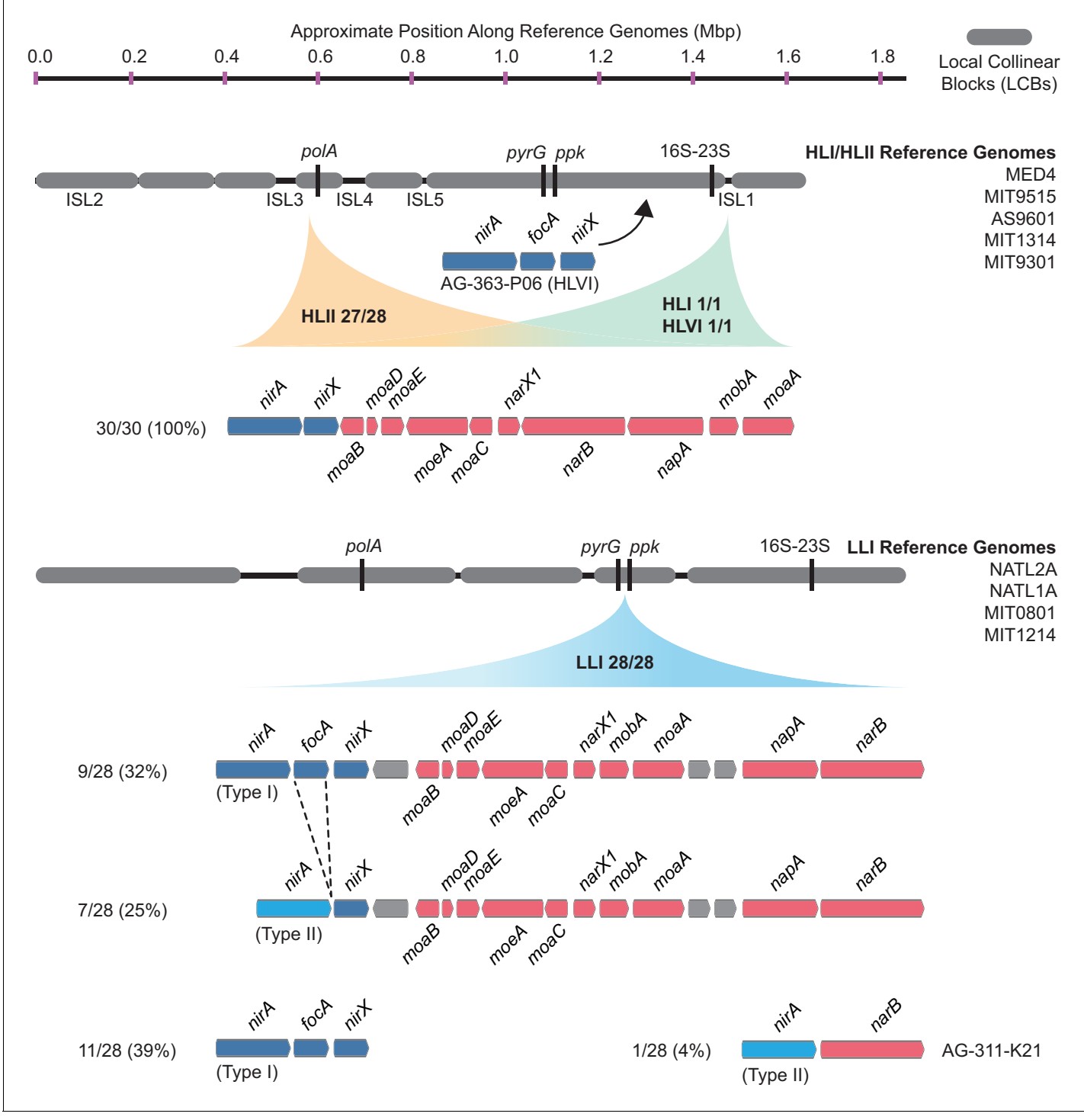

**Figure 4.** The gene order and genomic location of the nitrate assimilation gene cluster in high-light and low-light adapted *Prochlorococcus* genomes. The location of the nitrate and nitrite assimilation genes is shown relative to conserved core marker genes (black bars). The proportion of genomes in each clade with the indicated location is shown next to the clade names. The percentage of genomes in each group with a specific gene content and order is shown to the left of each gene order plot.

DOI: https://doi.org/10.7554/eLife.41043.011

The following figure supplements are available for figure 4:

**Figure supplement 1.** Mauve alignments visualized for representative contigs from HLII *Prochlorococcus* single cell genome assemblies in comparison to the reference genome *Prochlorococcus* AS9601 (HLII; non-nitrate assimilating).

*Figure 4 continued on next page*

*Figure 4 continued*

DOI: https://doi.org/10.7554/eLife.41043.012

**Figure supplement 2.** Mauve alignments visualized for representative contigs from HLI and HLVI *Prochlorococcus* single cell genome assemblies in comparison to the reference genomes *Prochlorococcus* MED4 (HLI) and *Prochlorococcus* AS9601 (HLII).

DOI: https://doi.org/10.7554/eLife.41043.013

**Figure supplement 3.** Mauve alignments visualized for representative contigs from LLI *Prochlorococcus* single cell genome assemblies in comparison to the reference genome *Prochlorococcus* NATL2A (LLI; nitrite assimilation only).

DOI: https://doi.org/10.7554/eLife.41043.014

recombination are likely a general feature of free-living marine bacterial populations (**Vergin et al., 2007**; **Vos and Didelot, 2009**), including *Prochlorococcus*.

To further evaluate these processes in *Prochlorococcus*, we next assessed the cross-habitat diversity of the nitrate assimilation genes in comparison to the core gene, *gyrB*, and the phosphate assimilation genes, *pstB* and *pstS*. The latter are examples of genes that are known to experience high rates of recombination, horizontal gene transfer, and/or efficient selection in *Prochlorococcus* populations (**Coleman and Chisholm, 2010**). Comparing single cells from two populations (North Pacific vs. North Atlantic), we found no significant phylogenetic divergence in the core gene *gyrB* and in genes in the upstream half of the nitrate assimilation pathway (**Table 2**, **Figure 5**). As expected, the *pstB* and *pstS* genes did cluster phylogenetically based on the populations from which they were derived (**Table 2**, **Figure 5**). Notably, the nitrite reductase gene *nirA* exhibited significant phylogenetic divergence between the two populations in 2 out of 3 subsampled data sets (**Table 2**). This gene may have experienced high enough recombination rates to facilitate its phylogenetic divergence between geographically distant populations or may have been fine-tuned by selection for optimal function in these contrasting ecosystems.

While it generally appears that recombination and selection have not been sufficient to drive significant phylogenetic divergence between populations for most nitrate assimilation genes (**Table 2**, **Figure 5**), we did observe a particularly high nucleotide similarity between these genes within clades

**Table 1.** Rates of recombination relative to mutation for representative genomic regions in high-light and low-light adapted *Prochlorococcus*.

| Region | Functional group | Sequences in alignment | Alignment length (bp) | $\kappa$ | $\delta$ | $\nu$ | $R/\theta$ | $\rho/\theta$ | $R/m$ |
|---|---|---|---|---|---|---|---|---|---|
| High-light adapted *Prochlorococcus* (HLII and HLVI clades) | | | | | | | | | |
| *nirA-moaA* | nitrate | 33 | 11428 | 3.32 | 4202 | 0.012 | 0.68 | 1.36 | 34.0 |
| *polA* region[*] | core flanking | 19 | 11554 | 3.72 | 8035 | 0.020 | 0.20 | 0.41 | 33.2 |
| Low-light adapted *Prochlorococcus* (LLI clade) | | | | | | | | | |
| *moaB-narB* | nitrate | 17 | 11985 | 2.91 | 435 | 0.021 | 1.78 | 3.57 | 16.5 |
| *pyrG* region[†] | core flanking | 15 | 10080 | 2.81 | 964 | 0.029 | 2.17 | 4.34 | 59.7 |
| *ppk* region[‡] | core flanking | 15 | 11918 | 3.19 | 1056 | 0.031 | 0.93 | 1.85 | 30.3 |

[*] 3' flanking region, containing *polA*, adjacent to the nitrate assimilation gene cluster in the HLII clade.

[†] 5' flanking region, containing *pyrG*, adjacent to the nitrate assimilation gene cluster in the LLI clade.

[‡] 3' flanking region, containing *ppk*, adjacent to the nitrate assimilation gene cluster in the LLI clade.

$\kappa$, Transition/transversion rate ratio as estimated by PhyML under the HKY85 model.

$\delta$, Mean length of DNA imported by homologous recombination.

$\nu$, Divergence rate, per site, of DNA imported by homologous recombination.

$R/\theta$, Per-site rate of initiation of recombination relative to the population mutation rate.

$\rho/\theta$, Population recombination rate relative to the population mutation rate ($\rho = 2R$).

r/m, Relative impact of recombination versus mutation on the per-site substitution rate. Equal to $(R/\theta) \times \delta \times \nu$.

DOI: https://doi.org/10.7554/eLife.41043.015

The following source data is available for Table 1:

Source data 1. Compressed tar archive (zip format) containing alignments used in CLONALFRAMEML analyses.

DOI: https://doi.org/10.7554/eLife.41043.016

**Table 2.** Significance (p value) for divergent phylogenetic gene clusters separating populations in the North Pacific Subtropical Gyre (HOT) and in the North Atlantic Subtropical Gyre (BATS).

| Gene | Product | Unifrac | P-test |
|------|---------|---------|--------|
| *amtB* | Ammonium transporter | 0.651, 0.135, 0.199 | 0.576, **0.006**, 0.554 |
| *glnA* | Glutamine synthetase | 0.058, 0.129, 0.136 | 0.045, 0.238, 0.214 |
| *glsF* | Glutamate synthase | 0.277, 0.066, 0.428 | 0.547, 0.235, 0.249 |
| *gyrB* | DNA gyrase subunit B | 0.425, 0.110, 0.477 | 0.867, 0.058, **0.006** |
| *moaA* | Molybdopterin biosynthesis | **0.040**, 0.138, 0.090 | 0.067, 0.577, 0.555 |
| *napA* | Nitrate/nitrite transporter | 0.427, 0.082, 0.600 | 0.229, 0.051, 0.225 |
| *narB* | Nitrate reductase | 0.326, 0.547, 0.355 | 0.558, 0.060, 0.221 |
| *nirA* | Nitrite reductase | **0.004**, **0.039**, 0.072 | **0.010**, 0.066, 0.056 |
| *pstB* | Phosphate transporter ATP-binding | **0.001**,**<0.001**,**<0.001** | **0.001**,**<0.001**,**<0.001** |
| *pstS* | Phosphate transporter substrate binding | **<0.001**,**<0.001**,**<0.001** | **<0.001**,**<0.001**,**<0.001** |

P values are reported for three replicate analyses using 18 single cells subsampled from HOT (AG-347; n = 9) and BATS (AG-355; n = 9). Trees exhibiting significant phylogenetic divergence between the two populations (p<0.05) are in bold face type.

DOI: https://doi.org/10.7554/eLife.41043.018

The following source data is available for Table 2:

**Source data 1.** Compressed tar archive (zip format) containing the alignments and group files used in beta diversity analyses.

DOI: https://doi.org/10.7554/eLife.41043.019

(*Figure 6*). Homologous recombination within the nitrate assimilation gene clusters of two closely related genomes can serve as a cohesive force that reinforces genetic similarity. Further, tests of adaptive evolution indicated that the nitrate assimilation genes are subject to strong purifying selection (*Table 3*, *Table 4*, *Table 5*) which would act to constrain divergence at non-synonymous sites.

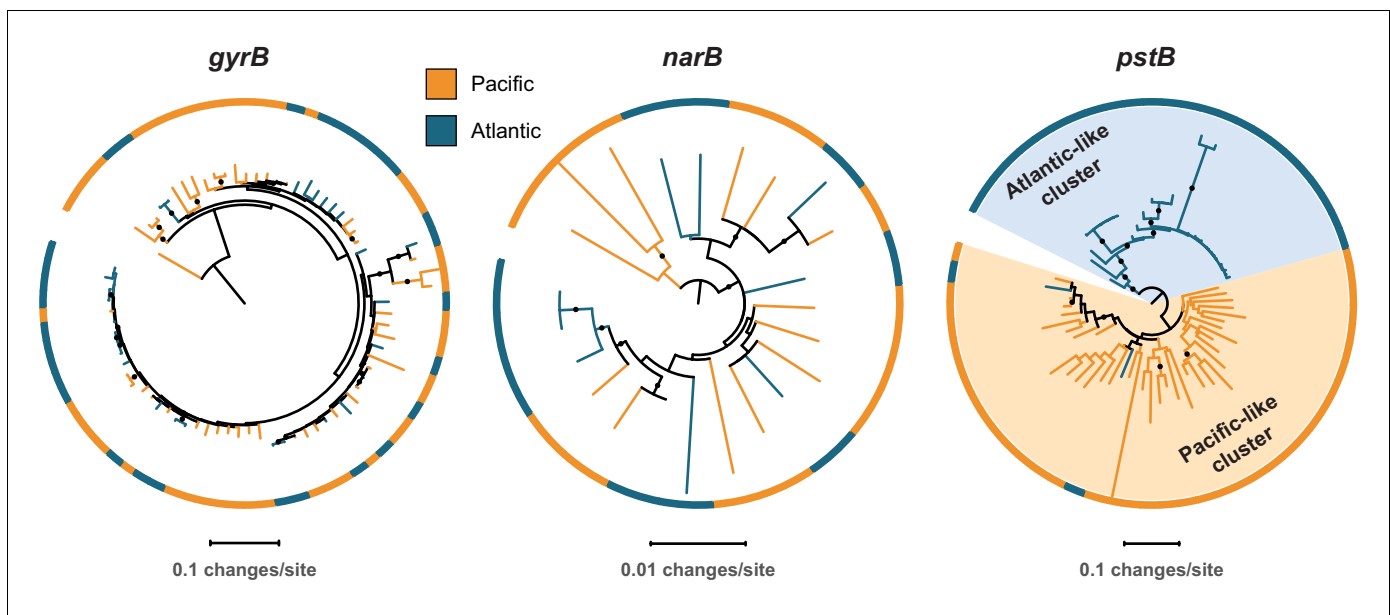

**Figure 5.** Representative phylogenetic patterns for DNA gyrase subunit B (*gyrB*) in comparison to nitrate reductase (*narB*) and the phosphate assimilation gene *pstB* (encoding the ABC transporter ATP binding subunit) from single cells in surface populations at HOT (Hawai'i Ocean Time-series) and BATS (Bermuda Atlantic Timeseries Study). The *gyrB* and *narB* genes do not exhibit significant phylogenetic divergence between the two sites. The *pstB* gene, in contrast, has significantly diverged into Atlantic-like and Pacific-like clusters of sequences due to frequent recombination, gene transfer, and/or selection (sensu *Coleman and Chisholm, 2010*).

DOI: https://doi.org/10.7554/eLife.41043.017

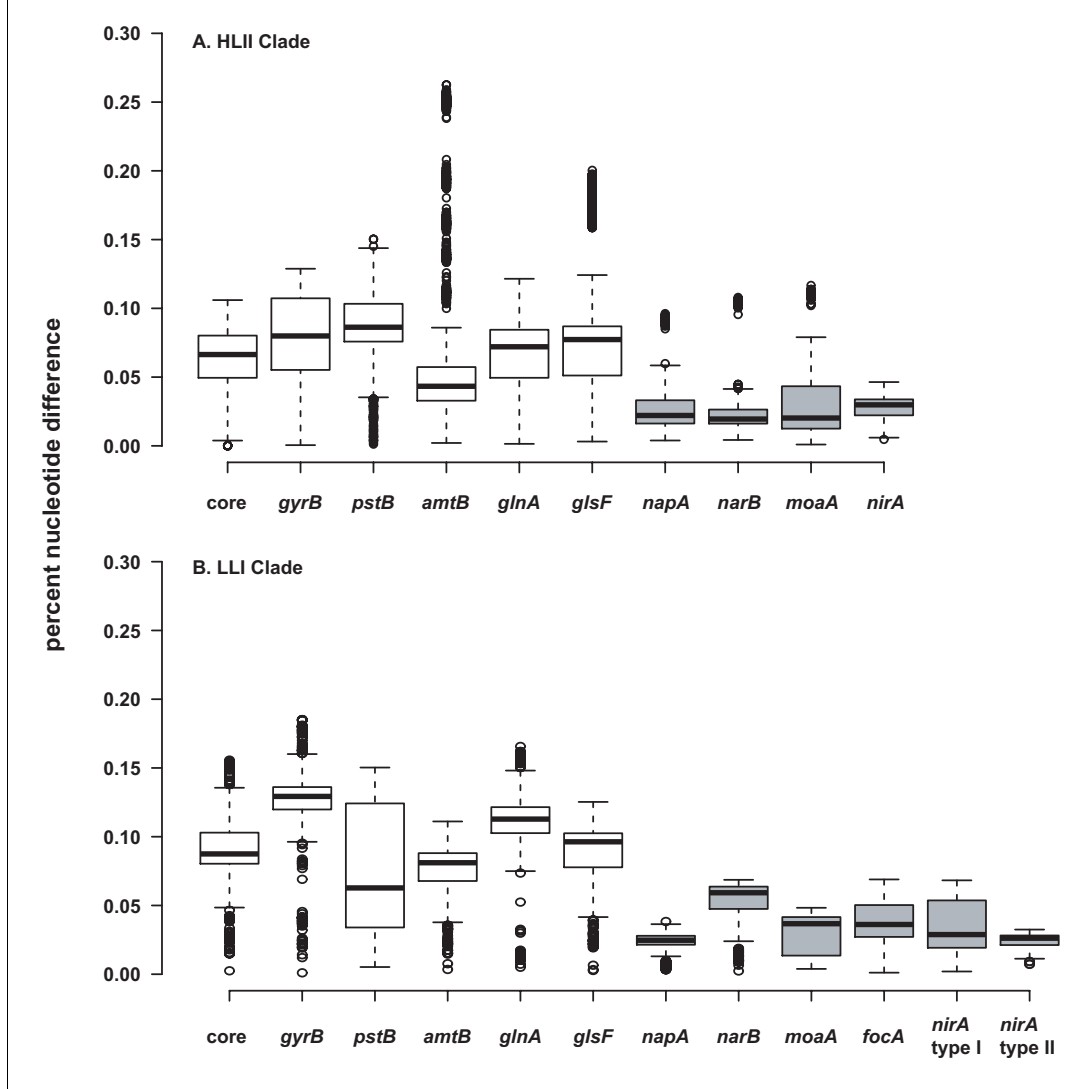

**Figure 6.** Percent nucleotide difference of selected genes for the HLII. (**A**) and LLI (**B**) clades of *Prochlorococcus*. The 'core' genes are a concatenated alignment of up to 37 PhyloSift marker genes. Genes in the nitrate assimilation cluster are in gray. Center lines show the medians, box limits indicate the 25th and 75th percentiles, whiskers extend 1.5 times the interquartile range from the 25th and 75th percentiles, and outliers are represented by dots. For HLII genes (**A**), n = 5151, 4186, 3321, 3570, 4371, 3240, 378, 351, 351, 351 sample points. For LLI genes (**B**), n = 496, 406, 435, 435, 406, 406, 153, 136, 136, 171, 210, 28 sample points.

DOI: https://doi.org/10.7554/eLife.41043.020

The following source data is available for figure 6:

**Source data 1.** Compressed tar archive (zip format) containing the alignments (fasta format) and column distance matrices used in the preparation of *Figure 6*.

DOI: https://doi.org/10.7554/eLife.41043.021

Homologous recombination between existing nitrate assimilation gene clusters is expected to have a complementary role in maintaining the observed phylogenetic cohesion within individual clades of *Prochlorococcus* (*Figure 6*).

Homologous recombination in shared genomic regions that flank the nitrate assimilation gene cluster has likely facilitated both the loss and acquisition of these genes within clades. Given that rates of loss are expected to be several orders of magnitude faster than those of gains (*Apagyi et al., 2018*), the patchy distribution of the nitrate assimilation trait across the phylogeny of individual *Prochlorococcus* clades is more likely dominated by gene loss. Low nucleotide diversity

**Table 3.** Estimates of dN/dS and results of site model tests for adaptive evolution among codon sites for *Prochlorococcus* and *Synechococcus*.

Bolded LRT statistic values are chi-square critical values that meet a significance level of <0.001. For all genes, the inclusion of a class of neutral sites (M1) fits the data better than one dN/dS value for all sites (M0). While the inclusion of a class of sites under positive selection may be statistically justified under the M2 and M8 models, all dN/dS values are well below one suggesting that most sites are under purifying or neutral selection.

| | | | log-likelihood of site models for adaptive evolution | | | | | likelihood ratio test (LRT) statistic for model pairs (degrees of freedom) | | |
|---|---|---|---|---|---|---|---|---|---|---|
| Gene | Taxa | dN/dS | M0 | M1 | M2 | M7 | M8 | M0 vs. M1 (1) | M1 vs. M2 (2) | M7 vs. M8 (2) |
| *gyrB* | 226 | 0.036 | −105998 | −101367 | −101367 | −102434 | −100472 | **9263** | 0 | **3924** |
| *pstB* | 200 | 0.045 | −39945 | −39466 | −39466 | −38543 | −38484 | **957** | 0 | **118** |
| *amtB* | 200 | 0.066 | −70319 | −68351 | −68234 | −68226 | −67620 | **3935** | **235** | **1212** |
| *glnA* | 229 | 0.031 | −66948 | −65905 | −65905 | −65462 | −65118 | **2086** | 0 | **688** |
| *glsF* | 215 | 0.103 | −267696 | −249591 | −249591 | −253085 | −246724 | **36210** | 0 | **12722** |
| *napA* | 76 | 0.054 | −23709 | −23028 | −23028 | −23201 | −22961 | **1363** | 0 | **480** |
| *narB* | 78 | 0.119 | −40791 | −38790 | −38790 | −39102 | −38498 | **4001** | 0 | **1209** |
| *moaA* | 67 | 0.206 | −24794 | −22973 | −22931 | −23268 | −22770 | **3643** | **84** | **996** |
| *focA* | 60 | 0.078 | −18477 | −17701 | −17701 | −17895 | −17615 | **1553** | 0 | **562** |
| *nirA* | 115 | 0.109 | −54829 | −52361 | −52361 | −52063 | −51470 | **4937** | 0 | **1187** |

DOI: https://doi.org/10.7554/eLife.41043.022

The following source data is available for Table 3:

Source data 1. Compressed tar archive (zip format) containing example codeml control files, codon alignments (phylip format), and tree files (newick format) used for site model tests of adaptive evolution.
DOI: https://doi.org/10.7554/eLife.41043.023

among nitrate assimilation genes within clades (*Figure 6*), also suggests that replacement and/or acquisition of alleles through homologous recombination has driven occasional sweeps of nitrate assimilation alleles through populations of closely related *Prochlorococcus*. Overall, we expect that the maintenance of this trait in wild populations is a consequence of the higher relative fitness of nitrate assimilating cells under conditions of overall nitrogen limitation (*Berube et al., 2016*). It is also likely that frequency-dependent selection processes (*Cordero and Polz, 2014*) have played a role in setting the equilibrium between genotypes that possess the nitrate assimilation genes and those that lack them.

## Macroevolutionary implications of differential trait variability in *Prochlorococcus* clades and ecotypes

Our gene content, phylogenomic, and diversity analyses suggest that the evolution of nitrate assimilation in *Prochlorococcus* largely proceeds through vertical descent, stochastic gene loss, and homologous recombination between closely related cells. However, this leaves several patterns unexplained. In the absence of horizontal gene transfer between distant taxa, how can a trait, that is present in recently branching lineages, be absent in more deeply branching lineages? We addressed this by looking to the broader physiological and ecological context of *Prochlorococcus'* evolution. Among high-light adapted *Prochlorococcus*, the temporal and spatial distributions of cells containing *narB* in the wild suggests that selection favors cells that can assimilate nitrate under conditions where the supply of light is enhanced and the overall supply of nitrogen is low (*Berube et al., 2016*). Although nitrate is often scarce at the surface of the oligotrophic ocean (*Gruber, 2008*), we hypothesize that access to a wider array of inorganic nitrogen species (including nitrate) is advantageous when cells are limited by the overall supply of nitrogen. With increasing depth, we expect selection pressures to differ due to decreasing photon flux density (*Clarke and James, 1939*) and increasing nitrogen availability (*Gruber, 2008*). In particular, cells belonging to the LLI clade are associated with the top of the nitracline, where nitrate concentrations begin to increase with depth

**Table 4.** Branch-site model tests for adaptive evolution among codon sites for the foreground HLII branch.

Background lineages include *Synechococcus* and all other *Prochlorococcus*. Bolded LRT statistic values are chi-square critical values that meet a significance level of <0.001 with 1 degree of freedom.

| Gene | Hypothesis log-likelihood | | Site class 0 | Site class 1 | Site class 2a | Site class 2b | LRT statistic |
|---|---|---|---|---|---|---|---|
| gyrB | H0 −104105 | proportion of sites | 0.98759 | 0.00138 | 0.01102 | 0.00002 | **6074** |
| | | background dN/dS | 0 | 1 | 0 | 1 | |
| | | foreground dN/dS | 0 | 1 | 1 | 1 | |
| | H1 −101068 | proportion of sites | 0.99666 | 0.00177 | 0.00157 | 0 | |
| | | background dN/dS | 0.00908 | 1 | 0.00908 | 1 | |
| | | foreground dN/dS | 0.00908 | 1 | 1 | 1 | |
| pstB | H0 −39463 | proportion of sites | 0.99801 | 0.00161 | 0.00038 | 0 | 0 |
| | | background dN/dS | 0.03255 | 1 | 0.03255 | 1 | |
| | | foreground dN/dS | 0.03255 | 1 | 1 | 1 | |
| | H1 −39463 | proportion of sites | 0.99801 | 0.00161 | 0.00038 | 0 | |
| | | background dN/dS | 0.03255 | 1 | 0.03255 | 1 | |
| | | foreground dN/dS | 0.03255 | 1 | 1 | 1 | |
| amtB | H0 −68231 | proportion of sites | 0.9926 | 0.00663 | 0.00076 | 0.00001 | −2 |
| | | background dN/dS | 0.02052 | 1 | 0.02052 | 1 | |
| | | foreground dN/dS | 0.02052 | 1 | 1 | 1 | |
| | H1 −68232 | proportion of sites | 0.99288 | 0.00657 | 0.00054 | 0 | |
| | | background dN/dS | 0.0206 | 1 | 0.0206 | 1 | |
| | | foreground dN/dS | 0.0206 | 1 | 2.17216 | 2.17216 | |
| glnA | H0 −65896 | proportion of sites | 0.99839 | 0.00144 | 0.00018 | 0 | 0 |
| | | background dN/dS | 0.01722 | 1 | 0.01722 | 1 | |
| | | foreground dN/dS | 0.01722 | 1 | 1 | 1 | |
| | H1 −65896 | proportion of sites | 0.99839 | 0.00144 | 0.00018 | 0 | |
| | | background dN/dS | 0.01722 | 1 | 0.01722 | 1 | |
| | | foreground dN/dS | 0.01722 | 1 | 1 | 1 | |
| glsF | H0 −249282 | proportion of sites | 0.99092 | 0.00679 | 0.00227 | 0.00002 | 0 |
| | | background dN/dS | 0.0227 | 1 | 0.0227 | 1 | |
| | | foreground dN/dS | 0.0227 | 1 | 1 | 1 | |
| | H1 −249282 | proportion of sites | 0.99092 | 0.00679 | 0.00227 | 0.00002 | |
| | | background dN/dS | 0.0227 | 1 | 0.0227 | 1 | |
| | | foreground dN/dS | 0.0227 | 1 | 1 | 1 | |
| napA | H0 −23009 | proportion of sites | 0.97966 | 0.01219 | 0.00805 | 0.0001 | 0 |
| | | background dN/dS | 0.00928 | 1 | 0.00928 | 1 | |
| | | foreground dN/dS | 0.00928 | 1 | 1 | 1 | |
| | H1 −23009 | proportion of sites | 0.97966 | 0.01219 | 0.00805 | 0.0001 | |
| | | background dN/dS | 0.00928 | 1 | 0.00928 | 1 | |
| | | foreground dN/dS | 0.00928 | 1 | 1 | 1 | |
| narB | H0 −38711 | proportion of sites | 0.94311 | 0.02164 | 0.03446 | 0.00079 | 0 |
| | | background dN/dS | 0.02489 | 1 | 0.02489 | 1 | |
| | | foreground dN/dS | 0.02489 | 1 | 1 | 1 | |
| | H1 −38711 | proportion of sites | 0.94311 | 0.02164 | 0.03447 | 0.00079 | |
| | | background dN/dS | 0.02489 | 1 | 0.02489 | 1 | |
| | | foreground dN/dS | 0.02489 | 1 | 1 | 1 | |

*Table 4 continued on next page*

*Table 4 continued*

| Gene | Hypothesis log-likelihood | | Site class 0 | Site class 1 | Site class 2a | Site class 2b | LRT statistic |
|------|---------------------------|--|--------------|--------------|---------------|---------------|---------------|
| *moaA* | H0 −22958 | proportion of sites | 0.94068 | 0.03375 | 0.02468 | 0.00089 | 0 |
| | | background dN/dS | 0.03882 | 1 | 0.03882 | 1 | |
| | | foreground dN/dS | 0.03882 | 1 | 1 | 1 | |
| | H1 −22958 | proportion of sites | 0.94068 | 0.03375 | 0.02468 | 0.00089 | |
| | | background dN/dS | 0.03882 | 1 | 0.03882 | 1 | |
| | | foreground dN/dS | 0.03882 | 1 | 1 | 1 | |
| *nirA* | H0 −52348 | proportion | 0.97996 | 0.01302 | 0.00693 | 0.00009 | 14 |
| | | background dN/dS | 0.04327 | 1 | 0.04327 | 1 | |
| | | foreground dN/dS | 0.04327 | 1 | 1 | 1 | |
| | H1 −52341 | proportion | 0.97254 | 0.0124 | 0.01487 | 0.00019 | |
| | | background dN/dS | 0.04293 | 1 | 0.04293 | 1 | |
| | | foreground dN/dS | 0.04293 | 1 | 1 | 1 | |

DOI: https://doi.org/10.7554/eLife.41043.024

The following source data is available for Table 4:

Source data 1. Compressed tar archive (zip format) containing example codeml control files, codon alignments (phylip format), and tree files (newick format) used for branch-site model tests of adaptive evolution in the HLII clade.

DOI: https://doi.org/10.7554/eLife.41043.025

(*Berube et al., 2016*). Here we hypothesize that competition with ammonia and nitrite oxidizing microorganisms for reduced nitrogen sources may favor retention of the nitrate assimilation trait in these *Prochlorococcus* (*Berube et al., 2016*). At even greater depths within the euphotic zone, low irradiance likely offsets any fitness benefit associated with nitrate assimilation, even when nitrate is abundant.

Since the trait appears to have descended vertically in *Prochlorococcus* as a whole, this raises the possibility that basal LL clades, which lack nitrate assimilation and dominate at depth in the contemporary oceans, are derived from ancestral populations that had a broader depth distribution as well as the ability to use nitrate (*Figure 7*, *Figure 7—figure supplement 1*). Loss of nitrate assimilation genes in basal LL clades, as they became restricted to the bottom of the euphotic zone, is consistent with the argument that the long-term evolution of *Prochlorococcus* proceeded via a sequence of niche-constructing, adaptive radiations. In this model of *Prochlorococcus'* evolution, metabolic innovations enhanced overall nutrient affinity by increasing the photosynthetic electron flux, thereby facilitating nutrient uptake at ever lower nutrient levels, making new realized niches available to *Prochlorococcus* and steadily narrowing the depth range of deeper branching clades (*Braakman et al., 2017*).

This model can be understood by considering the free energy cost of the reaction for nutrient uptake transport ($\Delta_r G$), which is given by $\Delta_r G = RT \ln([n]_I/[n]_E) + zF\Delta\Psi$, where $R$ is the gas constant, $T$ is the temperature, $[n]_I$ and $[n]_E$ are the concentrations of nutrient $n$ inside and outside the cell, respectively, $z$ is the unit charge of nutrient $n$, $F$ is the Faraday constant and $\Delta\Psi$ is the membrane potential. Thus, as environmental nutrient levels decrease, the free energy cost of nutrient uptake increases, linking nutrient affinity directly to the energy flux of cells. Evolutionary reconstructions based on a wide range of evidence from the metabolic network and photosynthetic machinery, ecology and light physiology in turn suggests that over the course of evolution *Prochlorococcus* cells have responded to this selective pressure by steadily increasing their electron-to-nutrient flux ratio ($\nu_e/\nu_n$), which enhances the transfer of solar energy into metabolism (*Braakman et al., 2017*). Thus, it is inferred that each new ecotype increased the harvesting of solar energy in the brightly illuminated but nutrient poor surface waters, thereby increasing the energetic driving of nutrient uptake and serially drawing down nutrient levels and restricting basal lineages to deeper waters (*Braakman et al., 2017*).

In our proposed macroevolutionary scenario, nitrate assimilation genes descend vertically from ancestral populations and stochastic loss of these genes is balanced by the fitness benefit of nitrate

**Table 5.** Branch-site model tests for adaptive evolution among codon sites for the foreground LLI branch.
Background lineages include *Synechococcus* and all other *Prochlorococcus*. Bolded LRT statistic values are chi-square critical values that meet a significance level of <0.001 with 1 degree of freedom.

| Gene | Hypothesis log-likelihood | | Site class 0 | Site class 1 | Site class 2a | Site class 2b | LRT statistic |
|---|---|---|---|---|---|---|---|
| *gyrB* | H0 −101367 | proportion of sites | 0.99813 | 0.00187 | 0 | 0 | **22** |
| | | background dN/dS | 0.01062 | 1 | 0.01062 | 1 | |
| | | foreground dN/dS | 0.01062 | 1 | 1 | 1 | |
| | H1 −101356 | proportion of sites | 0.99768 | 0.00186 | 0.00046 | 0 | |
| | | background dN/dS | 0.01053 | 1 | 0.01053 | 1 | |
| | | foreground dN/dS | 0.01053 | 1 | 1 | 1 | |
| *pstB* | H0 −39407 | proportion of sites | 0.98046 | 0.00147 | 0.01805 | 0.00003 | 0 |
| | | background dN/dS | 0.03172 | 1 | 0.03172 | 1 | |
| | | foreground dN/dS | 0.03172 | 1 | 1 | 1 | |
| | H1 −39407 | proportion of sites | 0.98046 | 0.00147 | 0.01805 | 0.00003 | |
| | | background dN/dS | 0.03172 | 1 | 0.03172 | 1 | |
| | | foreground dN/dS | 0.03172 | 1 | 1 | 1 | |
| *amtB* | H0 −68211 | proportion of sites | 0.99157 | 0.00673 | 0.00169 | 0.00001 | 0 |
| | | background dN/dS | 0.01992 | 1 | 0.01992 | 1 | |
| | | foreground dN/dS | 0.01992 | 1 | 1 | 1 | |
| | H1 −68211 | proportion of sites | 0.9918 | 0.00677 | 0.00143 | 0.00001 | |
| | | background dN/dS | 0.01992 | 1 | 0.01992 | 1 | |
| | | foreground dN/dS | 0.01992 | 1 | 1 | 1 | |
| *glnA* | H0 −65885 | proportion of sites | 0.99799 | 0.00141 | 0.0006 | 0 | 0 |
| | | background dN/dS | 0.01727 | 1 | 0.01727 | 1 | |
| | | foreground dN/dS | 0.01727 | 1 | 1 | 1 | |
| | H1 −65885 | proportion of sites | 0.99799 | 0.00141 | 0.0006 | 0 | |
| | | background dN/dS | 0.01727 | 1 | 0.01727 | 1 | |
| | | foreground dN/dS | 0.01727 | 1 | 1 | 1 | |
| *glsF* | H0 −249371 | proportion of sites | 0.98865 | 0.00686 | 0.00445 | 0.00003 | 0 |
| | | background dN/dS | 0.02306 | 1 | 0.02306 | 1 | |
| | | foreground dN/dS | 0.02306 | 1 | 1 | 1 | |
| | H1 −249371 | proportion of sites | 0.98865 | 0.00686 | 0.00445 | 0.00003 | |
| | | background dN/dS | 0.02306 | 1 | 0.02306 | 1 | |
| | | foreground dN/dS | 0.02306 | 1 | 1 | 1 | |
| *napA* | H0 −23014 | proportion of sites | 0.97128 | 0.01248 | 0.01604 | 0.00021 | 0 |
| | | background dN/dS | 0.01004 | 1 | 0.01004 | 1 | |
| | | foreground dN/dS | 0.01004 | 1 | 1 | 1 | |
| | H1 −23014 | proportion of sites | 0.97122 | 0.01246 | 0.01611 | 0.00021 | |
| | | background dN/dS | 0.01004 | 1 | 0.01004 | 1 | |
| | | foreground dN/dS | 0.01004 | 1 | 1 | 1 | |
| *narB* | H0 −38790 | proportion of sites | 0.97596 | 0.02404 | 0 | 0 | **40** |
| | | background dN/dS | 0.02939 | 1 | 0.02939 | 1 | |
| | | foreground dN/dS | 0.02939 | 1 | 1 | 1 | |
| | H1 −38770 | proportion of sites | 0.9577 | 0.02346 | 0.01839 | 0.00045 | |
| | | background dN/dS | 0.02762 | 1 | 0.02762 | 1 | |
| | | foreground dN/dS | 0.02762 | 1 | 1 | 1 | |

*Table 5 continued on next page*

*Table 5 continued*

| Gene | Hypothesis log-likelihood | | Site class 0 | Site class 1 | Site class 2a | Site class 2b | LRT statistic |
|---|---|---|---|---|---|---|---|
| *moaA* | H0 −22961 | proportion of sites | 0.90692 | 0.03565 | 0.05526 | 0.00217 | 0 |
| | | background dN/dS | 0.03704 | 1 | 0.03704 | 1 | |
| | | foreground dN/dS | 0.03704 | 1 | 1 | 1 | |
| | H1 −22961 | proportion of sites | 0.90692 | 0.03565 | 0.05526 | 0.00217 | |
| | | background dN/dS | 0.03704 | 1 | 0.03704 | 1 | |
| | | foreground dN/dS | 0.03704 | 1 | 1 | 1 | |
| *focA* | H0 −17695 | proportion of sites | 0.97605 | 0.01411 | 0.00969 | 0.00014 | 0 |
| | | background dN/dS | 0.01425 | 1 | 0.01425 | 1 | |
| | | foreground dN/dS | 0.01425 | 1 | 1 | 1 | |
| | H1 −17695 | proportion of sites | 0.97605 | 0.01411 | 0.0097 | 0.00014 | |
| | | background dN/dS | 0.01425 | 1 | 0.01425 | 1 | |
| | | foreground dN/dS | 0.01425 | 1 | 1 | 1 | |
| *nirA* type I | H0 −52349 | proportion of sites | 0.97292 | 0.01243 | 0.01446 | 0.00018 | 0 |
| | | background dN/dS | 0.04333 | 1 | 0.04333 | 1 | |
| | | foreground dN/dS | 0.04333 | 1 | 1 | 1 | |
| | H1 −52349 | proportion of sites | 0.97292 | 0.01243 | 0.01446 | 0.00018 | |
| | | background dN/dS | 0.04333 | 1 | 0.04333 | 1 | |
| | | foreground dN/dS | 0.04333 | 1 | 1 | 1 | |
| *nirA* type II | H0 −52358 | proportion of sites | 0.97139 | 0.01216 | 0.01625 | 0.0002 | 0 |
| | | background dN/dS | 0.0442 | 1 | 0.0442 | 1 | |
| | | foreground dN/dS | 0.0442 | 1 | 1 | 1 | |
| | H1 −52358 | proportion of sites | 0.97139 | 0.01216 | 0.01625 | 0.0002 | |
| | | background dN/dS | 0.0442 | 1 | 0.0442 | 1 | |
| | | foreground dN/dS | 0.0442 | 1 | 1 | 1 | |

DOI: https://doi.org/10.7554/eLife.41043.026

The following source data is available for Table 5:

Source data 1. Compressed tar archive (zip format) containing example codeml control files, codon alignments (phylip format), and tree files (newick format) used for branch-site model tests of adaptive evolution in the LLI clade.

DOI: https://doi.org/10.7554/eLife.41043.027

assimilation in environments where it offers a selective advantage. Ancestral state reconstruction suggests that this scenario is plausible when trait loss is favored over trait acquisition as suggested by the comparative genomics analysis of our data set (*Figure 7—figure supplement 1*). Over time, niche differentiation between *Prochlorococcus* ecotypes resulted in basal lineages losing nitrate assimilation genes as they became specialized to niches characterized by adaptation to lower irradiances (*Figure 7*). In the basal LLIV clade, which is restricted to the deepest region of the euphotic zone, loss of these genes appears to have run to completion rather than reaching some frequency-dependent equilibrium that is observed in the recently emerged clades. Homologous recombination is then generally limited to closely related cells which share sufficient nucleotide similarity to facilitate appreciable rates of replacement and/or acquisition of nitrate assimilation alleles (*Figure 7*).

While we cannot discount the possibility of non-homologous based gene acquisition in different genomic regions during speciation, the genomic rearrangement of nitrate assimilation genes in the most recently emerging lineages (*Figure 4*) is consistent with brief periods of enhanced genetic drift due to the founder effect (*Hallatschek et al., 2007*). Genomes within individual *Prochlorococcus* clades are highly syntenic, but large-scale genomic rearrangements exist between the genomes of different clades (*Yan et al., 2018*). Ecological differentiation of *Prochlorococcus* likely involved a period of rapid population expansion as new ecotypes gained access to a previously inaccessible nutrient pool (*Braakman et al., 2017*). Given the size of *Prochlorococcus* populations (*Biller et al.,*

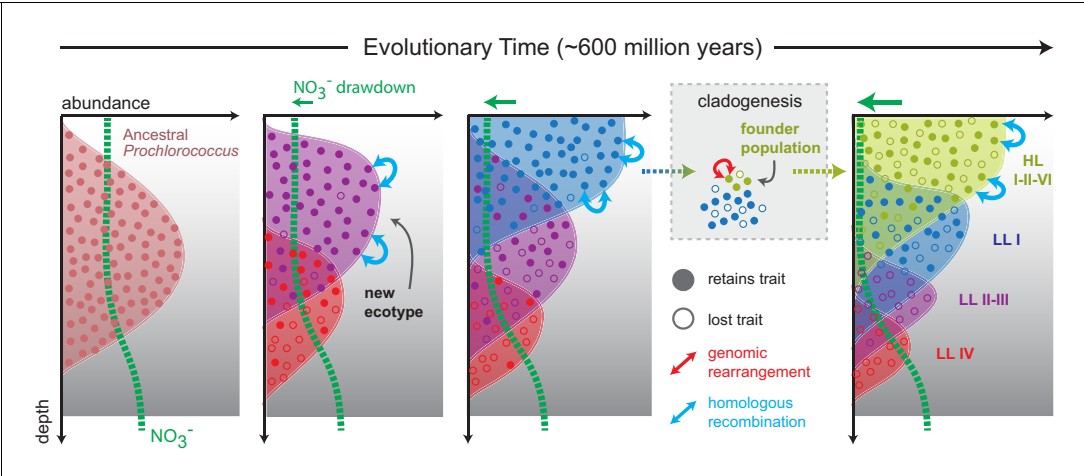

**Figure 7.** Proposed model mapping the vertical inheritance of the nitrate assimilation pathway onto a general model of speciation in *Prochlorococcus* (*Braakman et al., 2017*). We assume different cost:benefit ratios at top and bottom of the water column based on the energetic requirements of the nitrate assimilation pathway and the selective advantage of carrying this trait in environments with low nitrogen availability (see main text for further discussion). We further assume that ancestral *Prochlorococcus*, similar to *Synechococcus*, were capable of nitrate assimilation. As new Prochlorococcus clades/ecotypes emerged to more efficiently harness light energy and facilitate the draw-down of nitrogen at the surface, basal lineages were partitioned to the deeper regions of the euphotic zone (*Braakman et al., 2017*). In these basal lineages, higher relative costs of the pathway combined with access to other nitrogen sources (e.g. amino acids) hastened the loss of this trait through stochastic gene loss. In more recently emerging lineages (LLI, HLI, HLII, HLVI), the trait has been retained with intra-clade frequencies influenced by the specific chemical and physical characteristics of the environment in which they are found (*Berube et al., 2016*). The founder effect has driven punctuated changes (e.g. genome-wide rearrangements) during speciation, while homologous recombination has acted to constrain the divergence of gene sequence and order within clades.
DOI: https://doi.org/10.7554/eLife.41043.028

The following source data and figure supplement are available for figure 7:

**Source data 1.** Source data used to create *Figure 7—figure supplement 1*.
DOI: https://doi.org/10.7554/eLife.41043.030

**Figure supplement 1.** Reconstruction of the state of the ancestral *Prochlorococcus* node with regards to the presence or absence of *narB*.
DOI: https://doi.org/10.7554/eLife.41043.029

*2015*), any bottleneck at the front of this expansion is not expected to have been severe, but it has been observed that large genomic rearrangements can become fixed in bacterial populations even under weak bottlenecks and in the absence of horizontal gene transfer (*Raeside et al., 2014*). The increased likelihood of genome-wide genomic rearrangements following speciation events is a plausible explanation for the existence of nitrate assimilation gene clusters in different genomic locations when comparing between clades.

## Conclusions

We argue that the diversity and intraspecific distribution of the nitrate assimilation trait in *Prochlorococcus* is likely driven by a combination of vertical inheritance and gene loss, and rarely due to horizontal gene acquisition driven by non-homologous recombination. These processes have largely resulted in a patchy distribution of this trait among cells within clades that harbor nitrate assimilation genes. Homologous recombination further acts to constrain the divergence of this trait within clades and may have promoted gene-specific selective sweeps within populations of closely related cells. Seasonality in environmental conditions, under which this trait likely confers a selective advantage to cells (*Berube et al., 2016*), has probably facilitated the maintenance of this patchy trait distribution. We further propose that speciation and niche partitioning involved brief periods of enhanced drift and rapid change that facilitated the genomic rearrangement of nitrate assimilation genes. The restriction of basal *Prochlorococcus* lineages to greater depths during the general process of niche partitioning explains their loss of the nitrate assimilation trait (*Figure 7*). Overall, the underlying dynamics governing the loss and retention of nitrate assimilation within clades and during the genesis of new clades are closely intertwined. Superficially, this emergent pattern in microbial trait

variability might appear to be the result of horizontal gene transfer, but our evidence indicates that the observed patterns can be attributed to processes of vertical descent, gene loss, and recombination between close relatives that have operated throughout the entire radiation of *Prochlorococcus*.

## Materials and methods

### Dataset

A set of 463 *Prochlorococcus* and *Synechococcus* single cell genome assemblies (*Berube et al., 2018*) were filtered to exclude 76 genomes with less than 25% genome recovery, as determined by checkM (*Parks et al., 2015*). The AG-363-L17, AG-418-C09, and AG-418-C17 single cell genome assemblies were excluded because they contained both bacterial and phage genomes. A subset of 206 of the remaining single cells were screened using a PCR assay (detailed below) to confirm the presence or absence of *narB* in the amplified single cell DNA. An additional 33 single cell genome assemblies had annotated *narB* genes, but were not screened by PCR. This resulted in a total of 239 single cells with empirical evidence for the presence or absence of *narB,* and were retained in the data set; these single cell genome assemblies had a median genome recovery of 79%. The final data set encompassed 321 genomes after inclusion of reference culture genomes for 61 *Prochlorococcus* (26 closed and 35 permanent draft) and 21 *Synechococcus* (14 closed and seven permanent draft). Eight single cells, belonging to the HLIII and HLIV clades of *Prochlorococcus* (*Malmstrom et al., 2013*), were additionally used as references for the core marker gene phylogeny (*Figure 1*). All genomes were downloaded from the Integrated Microbial Genomes (IMG) system (*Chen et al., 2017*). ProPortal CyCOGs v6.0 definitions (*Berube et al., 2018*) were used as primary annotations. IMG accession numbers for the 321 genomes in the final data set are provided in *Supplementary file 1*.

### Genome sequencing

Genomes for the following cultured LLI *Prochlorococcus* strains were sequenced as part of this study: MIT0912, MIT0913, MIT0915, and MIT0917. The MIT0912 and MIT0913 strains were isolated in July 2009 from Station ALOHA (Hawai'i Ocean Time-series, 23.75°N, −158 °E) from a depth of 175 m on the HOT212 cruise (KM0915). MIT0915 and MIT0917 were derived from the P0902-H212 and P0903-H212 enrichment cultures, respectively (*Berube et al., 2016*). Cells were grown to mid exponential phase and pelleted by centrifugation. DNA was isolated by phenol/chloroform extraction (*Wilson, 2001*). PacBio library preparation and sequencing was carried out by the MIT BioMicro Center and the UMass Worcester Medical School's Deep Sequencing Core Facility. Assembly of PacBio reads was performed using the hierarchical genome-assembly process (Protocol = RS_HGAP_Assembly.2) as implemented in SMRT Analysis 2.3.0 (*Chin et al., 2013*) with the following parameters adjusted: Minimum Polymerase Read Quality = 0.85 and Genome Size = 2000000 bp (default settings were used for all other parameters). Overlapping ends of the assembly were identified using BLAST and the assembly was manually circularized. Circular assemblies were corrected using the RS_Resequencing.1 protocol in SMRT Analysis 2.3.0 (*Chin et al., 2013*) with the following parameters: Minimum Polymerase Read Quality = 0.85 and Consensus Algorithm = Quiver. These genomes were deposited with IMG (accession numbers: 2681812899, 2681812900, 2681812901, 2681812859), annotated using IMG Annotation Pipeline version 4 (*Markowitz et al., 2014*; *Chen et al., 2017*), and included in ProPortal CyCOGs v6.0 (*Berube et al., 2018*).

### Screening assay for *narB*

We used a PCR assay to screen for the presence/absence of the *narB* gene in the amplified DNA from sorted single cells. The design of the PCR primers was based on an alignment of 49 *narB* sequences extracted from 13 marine *Synechococcus* genomes (WH7805, WH7803, BL107, CC9902, CB0205, WH8102, CC9605, CB0101, RS9916, WH8016, CC9311, RCC307, and WH8109), 1 *Prochlorococcus* contig derived from an environmental metagenome assembly (*Astorga-Eló et al., 2015*), 11 LLI *Prochlorococcus* genomes (PAC1, MIT0915, MIT0917, AG-363-M20, AG-402-L20, AG-402-M23, AG-402-O21, AG-311-K16, AG-402-L09, AG-341-O20, and AG-331-D10), 1 HLII *Prochlorococcus* genome (AG-335-O19), 22 HLII *Prochlorococcus* genomes (SB, MIT0604, AG-355-I20, AG-402-I23,

AG-347-I22, AG-335-I15, AG-347-G18, AG-355-G23, AG-347-M23, AG-347-O22, AG-347-I19, AG-347-I21, AG-355-N18, AG-402-K16, AG-347-G20, AG-347-K17, AG-347-L20, AG-355-B23, AG-355-J09, AG-402-L23, AG-402-G23, and AG-402-K22), and 1 HLVI *Prochlorococcus* genome (AG-363-B18). Sequences were aligned by codon using MACSE (*Ranwez et al., 2011*) and the following primers were selected: narB.SAG.Deg.1786F (5'-CANTGGCAYACNATGAC-3') and narB.SAG.Deg.2004R (5'-RAANCCCCARTGCATNGG-3'). The forward primer has a 32-fold degeneracy targeting a conserved region encoding the [Q/H]WHTMT polypeptide, and the reverse primer has a 64-fold degeneracy targeting a conserved region encoding the PMHWGF polypeptide. These motifs are conserved between the *Prochlorococcus* and marine *Synechococcus* genomes in our data set and are expected to amplify most *narB* sequences within these groups. PCR conditions were optimized using *Prochlorococcus* strains SB and MIT0917. The reaction consisted of 1 μM forward primer, 1 μM reverse primer, 2 μl of template DNA, 200 μM dNTPs, 1x Phusion HF Buffer (New England BioLabs), one unit Phusion High-Fidelity DNA Polymerase (New England BioLabs), and 0.2x SYBR Green (Lonza) in a 30 μl total volume. Real-time PCR was performed using a Bio-Rad CFX96 instrument programmed for 40 cycles of 98°C for 10 s, 54°C for 30 s, and 72°C for 30 s. Positive amplification was identified by the presence of a single peak in the melting curve between 78–83°C and by visual inspection for an amplicon length of 219 bp on an agarose gel.

While PCR can identify genomic regions with poor sequencing coverage, biases in DNA amplification can result in gene copy numbers that still fall below the detection limit of PCR. Thus, we assessed false negative rates using a set of 4 reference cultures: *Prochlorococcus* SB (HLII clade, narB$^+$), *Prochlorococcus* MIT9301 (HLII clade, narB$^-$), *Prochlorococcus* MIT0917 (LLI clade, narB$^+$), and *Prochlorococcus* NATL2A (LLI clade, narB$^-$). Each culture was grown to mid-exponential phase, at which point, cell concentrations were determined by flow cytometry using a GUAVA flow cytometer (Millipore). The cultures were combined at equivalent concentrations, preserved using 10% glycerol, flash frozen in liquid nitrogen, and stored at −80°C. Single cells were sorted and the DNA was amplified using the WGA-X method (*Stepanauskas et al., 2017*) by the Bigelow Laboratory Single Cell Genomics Center. ITS sequences were amplified and sequenced using the Sanger method to associate wells on the 384 well plate with each control strain. The *narB* PCR assay was executed on the amplified single cell DNA from the control strains. Template DNA was prepared by making a 50-fold dilution of a subsample of the amplified DNA in 1x TE buffer. The false negative rate (likely due to extremely poor genome amplification across the nitrate assimilation gene cluster) was 8% for the control strains containing *narB*. We did not observe any false positive results for control strains lacking *narB*.

Our *narB* PCR assay was then applied to six experimental plates of amplified single cell DNA: AG-347, AG-355, AG-363, AG-402, AG-418, and AG-459. A SAG was deemed positive if PCR resulted in a narrow peak in the melting curve at approximately 78–83°C. Those with no amplification or with a melting curve characteristic of primer dimers were deemed negative. Twelve amplicons that were either borderline or potentially spurious were sequenced using the Sanger method to confirm whether or not they were *narB*. Finally, the results of the PCR screen were used to select an additional 47 narB$^+$ SAGs for whole genome sequencing, annotation by IMG (*Markowitz et al., 2014*), and inclusion in ProPortal CyCOGs v6.0 (*Berube et al., 2018*).

Comparison of PCR screening results with CyCOG annotations for the partial single cell genome assemblies supports the effectiveness of PCR as a method to assess the presence/absence of functional genes in large single cell genomics data sets. Of the 206 single cells screened by PCR, the PCR data and the annotation data agreed in 88% of cases. For single cells with an annotated *narB* gene (61/206 cells), 97% tested positive for the presence of *narB* by PCR (59/61 cells). For single cells that tested positive for the presence *narB* by PCR (81/206), 73% had an annotated *narB* gene (59/81 cells). The median genome recovery of these 81 single cell genomes was 80%, providing an expected occurrence of 65 single cell genomes that should contain an annotated *narB* gene. A post hoc analysis comparing 59 observed to 65 expected occurrences of genomes containing an annotated *narB* gene suggests an 9% false negative rate for the PCR screening assay relative to the 8% false negative rate determined using control strains.

## Phylogenetic inference

For core gene/protein phylogenies, we used PhyloSift (*Darling et al., 2014*) to search the genome assemblies in our data set for 37 marker gene families (*Darling et al., 2014*) and generate

concatenated nucleotide codon alignments for these marker genes. The alignments were manually curated to remove poorly aligned regions and translated to generate amino acid alignments. Maximum likelihood trees were then generated using RAxML 8.2.9 (*Stamatakis, 2014*) with automatic bootstopping criteria enabled and using the following command line parameters: raxmlHPC-PTHREADS-AVX -T 20 f a -N autoMRE -m PROTGAMMAAUTO (amino acid alignments); raxmlHPC-PTHREADS-AVX -T 20 f a -N autoMRE -m GTRCAT (nucleotide alignments). Trees were annotated using the Interactive Tree Of Life (iTOL) (*Letunic and Bork, 2016*).

For individual genes belonging to specific CyCOG families, we used MACSE (*Ranwez et al., 2011*) to generate nucleotide codon alignments. The alignments were manually curated to remove taxa with >80% missing data. For taxa with genes breaking across contig boundaries, the longest aligning segment was retained. The nucleotide alignments were translated to generate amino acid alignments and maximum likelihood trees were generated with RAxML 8.2.9 (*Stamatakis, 2014*), using and the command line parameters described for core protein marker phylogenies.

## Analysis of covariation in gene content

HLII and LLI single cells were filtered to exclude those with <75% genome recovery, as determined by checkM (*Parks et al., 2015*), yielding 105 single cells (83 HLII and 22 LLI single cells). Each group of genomes had median genome recoveries of 90% and 87%, respectively. The number of each CyCOG in each genome was enumerated and both abundance and binary (presence/absence) matrices were created for each set of HLII and LLI genomes. CyCOGs either shared by all genomes or exhibiting low representation among genomes have little information content with regards to the identification of co-varying genes. Thus, core CyCOGs and CyCOGs found in fewer than three genomes were excluded from analysis. Core CyCOGs in partial single cell genome assemblies were operationally defined as CyCOGs found in at least M% of genomes, where M% is the median genome recovery for each set of genomes (for example, CyCOGs found in at least 19 (22 × 0.87) LLI genomes were estimated to be core CyCOGs). Binary matrices were imported into MORPHEUS (Broad Institute; MA, USA, accessed at https://software.broadinstitute.org/morpheus/) and CyCOGs were hierarchically clustered based on the Jaccard distance measure and using average linkage to identify CyCOGs that co-vary with the nitrate assimilation gene cluster. Gene enrichment analysis was also used to examine covariation of CyCOGs with the nitrate assimilation genes. Over- and under-representation of CyCOGs were evaluated using BiNGO 3.0.3 (*Maere et al., 2005*) in Cytoscape 3.4 (*Shannon et al., 2003*). For each set of 83 HLII and 22 LLI single cells, genes found in genomes containing *narB* were evaluated against the set of genes found in all genomes in the set. Significant enrichment of flexible genes in the test set was assessed using the hypergeometric statistical test and the Benjamini and Hochberg correction for multiple hypothesis testing.

## Genome location and synteny

Single cell genome assemblies were filtered for contigs greater than 40 kbp in length that contained complete nitrite and/or nitrate assimilation gene clusters. This yielded contigs with sequence data extending from the nitrate assimilation gene cluster into adjacent genomic regions for 1 HLI, 26 HLII, 2 HLVI, and 18 LLI single cell genomes. Two HLII and 10 LLI culture genomes with nitrite and/or nitrate assimilation gene clusters were also included. Contigs were then aligned to closed reference genomes using the progressiveMauve algorithm (*Darling et al., 2010*) in MAUVE 2.4.0 with default parameters. Alignments for contigs derived from high-light adapted single cells used the following reference genomes: MIT9301 (HLII), AS9601 (HLII), MED4 (HLI), and MIT9515 (HLI). Alignments for contigs derived from low-light adapted single cells used the following reference genomes for the LLI clade: NATL2A, MIT0915, and MIT0917.

## Genetic distance of representative genes

Percent nucleotide difference for genes belonging to HLII and LLI taxa with at least 75% genome recovery were determined using MOTHUR (*Schloss et al., 2009*). Genes belonging to these genomes were extracted from the codon alignments and the dist.seqs command in MOTHUR (*Schloss et al., 2009*) was used to create a column distance matrix for each gene. BoxPlotR (*Spitzer et al., 2014*) was used for visualization.

## Recombination detection

Complete nitrate assimilation gene clusters and core flanking regions adjacent to the nitrate assimilation gene clusters were extracted from Mauve alignments. Nucleotide alignments for these genomic tracts were manually curated and PhyML 3.0 (*Guindon et al., 2010*) was used to construct a starting tree and to estimate the transition/transversion ratio (kappa) using default parameters. The impact of recombination relative to mutation was then assessed with CLONALFRAMEML (*Didelot and Wilson, 2015*) using the nucleotide alignment, phylogenetic tree, and estimated kappa value as inputs. The ρ/θ value is a measure of the occurrence of initiation or termination of recombination relative to the population mutation rate. The relative impact of recombination versus mutation (r/m) is measured by accounting for the length (δ) and genetic distance (ν) of the recombining fragments (*Didelot and Wilson, 2015*).

## Beta diversity analysis

Gene sequences belonging to HLII clade single cells derived from two surface populations (AG-347, HOT, Hawai'i Ocean Time-series, 5 m depth; and AG-355, BATS, Bermuda Atlantic Time-series Study, 10 m depth) were extracted from gene alignments. These two environmental samples had the highest sample number of HLII clade single cells and included a minimum of 9 sequence representatives for each population. Nine sequences were subsampled without replacement from each population using MOTHUR (*Schloss et al., 2009*) to yield a total of 18 sequences in the final alignment. The subsampled alignment was used to build a phylogenetic tree using RAxML 8.2.9 (*Stamatakis, 2014*) with the GTRCAT model and automatic bootstopping criteria enabled. Trees were rooted at the midpoint using PyCogent (*Knight et al., 2007*). Beta diversity was assessed using Fast Unifrac (*Hamady et al., 2010*) as implemented in PyCogent (*Knight et al., 2007*) to determine significance values based on Unifrac and the P-test (*Martin, 2002*): fast_unifrac_permutations_file (tree_in, envs_in, weighted = False, num_iters = 1000, test_on='Pairwise'); fast_p_test_file(tree_in, envs_in, num_iters = 1000, test_on='Pairwise'). This analysis was repeated for three independent subsampled data sets.

## Estimates of dN/dS and tests of adaptive evolution

The CODEML application of the PAML package (*Yang, 1997*; *Yang, 2007*) was used to estimate synonymous and nonsynonymous substitution rates (dN/dS) and assess the likelihood of adaptive evolution for a selection of protein coding genes (*gyrB*; *pstB*; *amtB*; *glnA*; *glsf*; *napA*; *narB*; *moaA*; *focA*; *nirA*). The CODEML parameters employed were as described by *Jeffares et al. (2015)* except for fixing branch lengths (fix_blength = 2) and using a small difference value (Small_Diff) of 1e-7 in site model tests and 1e-8 in branch-site model tests. Initial data sets were the codon alignments used for the gene and protein phylogenies described in phylogenetic inference section of the methods. Given the large number of taxa in the codon alignments and the high computational demands of CODEML, we found it necessary to identify representative taxa using MOTHUR (*Schloss et al., 2009*) by clustering the sequences with a distance cutoff of 0.01 using the following commands:

mothur unique.seqs(fasta = alignedcodons.fasta);

mothur dist.seqs(fasta = alignedcodons.unique.fasta);

mothur cluster(column = alignedcodons.unique.dist, name = alignedcodons.names, method = opti, cutoff = 0.01);

mothur get.oturep(column = alignedcodons.unique.dist, name = alignedcodons.names, list = alignedcodons.unique.opti_mcc.list)

The representative taxa identified by MOTHUR were extracted from the original codon alignments and RAxML (*Stamatakis, 2014*) was used to infer the phylogeny with the following parameters: raxmlHPC-PTHREADS-AVX -T 20 f a -x -p -N 100 m GTRCAT. Site model tests were run using the CODEML control file in *Table 3—source data 1*. dN/dS (omega) values and log-likelihoods for each of the five models were extracted from the output files. Significance was assessed using the likelihood ratio test statistic and the chi-squared distribution (*Jeffares et al., 2015*).

Branch-site model tests were run using the CODEML control files in *Table 4—source data 1* and *Table 5—source data 1* to assess the null hypothesis of adaptive evolution across sites and lineages (Hypothesis H0) with respect to the hypothesis for positive selection (Hypothesis H1), which allowed for a class of sites in the foreground branch with dN/dS > 1. The branch leading to the lineage under

examination (foreground branch) was marked with '$1' in the RAxML bestree file. dN/dS values for each site class in the foreground and background branches as well as log-likelihoods for the two hypotheses (H0 vs. H1) were extracted from their respective output files. Significance was assessed using the likelihood ratio test statistic and the chi-squared distribution. Although the likelihood ratio test does not follow the chi-squared distribution in the case of branch-site tests of adaptive evolution, the chi-squared critical value for 1 degree of freedom was used to make the test conservative (*Jeffares et al., 2015*).

## Ancestral state reconstruction

Mesquite 3.5.1 (*Maddison and Maddison, 2018*) was used to reconstruct the ancestral states for the nitrate assimilation trait using parsimony and maximum likelihood (*Maddison and Maddison, 2006*). All ancestral state reconstructions were based on the core marker protein phylogeny (*Figure 3*, panel a) which contained 321 taxa. The character states of extant taxa were a composite of the data compiled for the presence/absence of *narB* based on genome annotation and PCR screening. Two separate parsimony analyses were conducted, one in which gains and losses were equally weighted, and one in which the cost of gains was weighted 10 times higher than the cost of losses. For likelihood analyses, we compared a one-parameter Markov k-state (MK1) model with an asymmetrical 2-parameter Markov k-state model. The use of the asymmetrical model, which estimates the rates of gains and losses in the data set, was justified based on the likelihood ratio test statistic (LTR = 11.5) and the chi-squared distribution with 1 degree of freedom ($p < 0.0007$). The estimated forward (trait gain) rate was 14.5 and the estimated backward (trait loss) rate was 27.4 in the asymmetrical model.

## Acknowledgements

This work was supported by grants from the National Science Foundation (OCE-1153588 and DBI-0424599 to SWC and OCE-1335810 to RS), the Simons Foundation (Life Sciences Project Award IDs 337262 and 509034SCFY17, SWC; SCOPE Award ID 329108, SWC), and the Gordon and Betty Moore Foundation (Grant IDs GBMF495 and GBMF4511 to SWC). We thank Otto X Cordero (MIT) and Martin F Polz (MIT) for thoughtful discussions. This paper is a contribution from the Simons Collaboration on Ocean Processes and Ecology (SCOPE) and from the NSF Center for Microbial Oceanography: Research and Education (C-MORE).

## Additional information

### Funding

| Funder | Grant reference number | Author |
| --- | --- | --- |
| National Science Foundation | OCE-1153588 | Sallie W Chisholm |
| National Science Foundation | DBI-0424599 | Sallie W Chisholm |
| National Science Foundation | OCE-1335810 | Ramunas Stepanauskas |
| Simons Foundation | 337262 | Sallie W Chisholm |
| Simons Foundation | 329108 | Sallie W Chisholm |
| Gordon and Betty Moore Foundation | GBMF495 | Sallie W Chisholm |
| Gordon and Betty Moore Foundation | GBMF4511 | Sallie W Chisholm |
| Simons Foundation | 509034SCFY17 | Sallie W Chisholm |

The funders had no role in study design, data collection and interpretation, or the decision to submit the work for publication.

## Author contributions
Paul M Berube, Conceptualization, Data curation, Formal analysis, Supervision, Validation, Investigation, Visualization, Methodology, Writing—original draft, Project administration, Writing—review and editing, Designed the PCR screen and developed the macroevolution model; Anna Rasmussen, Validation, Investigation, Methodology, Writing—review and editing, Designed and performed the PCR screen; Rogier Braakman, Visualization, Writing—review and editing, Developed the macroevolution model; Ramunas Stepanauskas, Resources, Funding acquisition, Validation, Writing—review and editing; Sallie W Chisholm, Supervision, Funding acquisition, Writing—original draft, Project administration, Writing—review and editing

## Author ORCIDs
Paul M Berube (iD) http://orcid.org/0000-0001-5598-6602
Anna Rasmussen (iD) https://orcid.org/0000-0002-0031-2835
Ramunas Stepanauskas (iD) http://orcid.org/0000-0003-4458-3108

## Decision letter and Author response
Decision letter https://doi.org/10.7554/eLife.41043.044
Author response https://doi.org/10.7554/eLife.41043.045

# Additional files

## Supplementary files
• Supplementary file 1. Tab-delimited table containing the genomes and associated IMG accession numbers used in the final data set. Clade indicates the respective *Synechococcus* subcluster or *Prochlorococcus* clade. Results from the *narB* PCR screening assay are presented as a binary (0 = negative; 1 = positive; n.d = not determined). Estimated genome recovery was determined using checkM (*Parks et al., 2015*) and the presence/absence of annotated reductase and transporter genes for nitrite (*nirA* and *focA*) and nitrate (*narB* and *napA*) in each genome assembly are as given as a binary (0 = absent; 1 = present).
DOI: https://doi.org/10.7554/eLife.41043.031

• Transparent reporting form
DOI: https://doi.org/10.7554/eLife.41043.032

## Data availability
Sequencing data have been deposited in Integrated Microbial Genomes under the accession numbers (IMG Genome ID) 2681812899, 2681812900, 2681812901, and 2681812859. All data generated or analyzed during this study are included in the manuscript and supporting files. Source data files have been provided for Figure 1, Figure 2, Figure 3, Figure 6, Figure 7, Table 1, Table 2, Table 3, Table 4, and Table 5.

The following datasets were generated:

| Author(s) | Year | Dataset title | Dataset URL | Database and Identifier |
|---|---|---|---|---|
| Berube PM, Chisholm SW | 2018 | Prochlorococcus sp. MIT0912 | https://img.jgi.doe.gov/cgi-bin/m/main.cgi?section=TaxonDetail&page=taxonDetail&taxon_oid=2681812899 | Integrated Microbial Genomes & Microbiomes, 2681812899 |
| Berube PM, Chisholm SW | 2018 | Prochlorococcus sp. MIT0913 | https://img.jgi.doe.gov/cgi-bin/m/main.cgi?section=TaxonDetail&page=taxonDetail&taxon_oid=2681812900 | Integrated Microbial Genomes & Microbiomes, 2681812900 |
| Berube PM, Chisholm SW | 2018 | Prochlorococcus sp. MIT0915 | https://img.jgi.doe.gov/cgi-bin/m/main.cgi?section=TaxonDetail&page=taxonDetail&taxon_oid=2681812901 | Integrated Microbial Genomes & Microbiomes, 2681812901 |

| | | | | |
|---|---|---|---|---|
| Berube PM, Chisholm SW | 2018 | Prochlorococcus sp. MIT0917 | https://img.jgi.doe.gov/cgi-bin/m/main.cgi?section=TaxonDetail&page=taxonDetail&taxon_oid=2681812859 | Integrated Microbial Genomes & Microbiomes, 2681812859 |

The following previously published dataset was used:

| Author(s) | Year | Dataset title | Dataset URL | Database and Identifier |
|---|---|---|---|---|
| Berube PM, Biller SJ, Hackl T, Hogle SL, Satinsky BM, Becker JW, Braakman R, Collins SB, Kelly L, Berta-Thompson J, Coe A, Bergauer K, Bouman HA, Browning TJ, De Corte D, Hassler C, Hulata Y, Jacquot JE, Maas EW, Reinthaler T, Sintes E | 2018 | Single cell genomes of Prochlorococcus, Synechococcus, and sympatric microbes from diverse marine environments | https://www.ncbi.nlm.nih.gov/bioproject/PRJNA445865/ | NCBI BioProject, BioProject PRJNA445865 |

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
