## [Decision Letter]

Thank you for submitting your article "Emergence of trait variability through the lens of nitrogen assimilation in *Prochlorococcus*" for consideration by *eLife*. Your article has been reviewed by two peer reviewers, and the evaluation has been overseen by a Reviewing Editor and Ian Baldwin as the Senior Editor. The following individual involved in review of your submission has agreed to reveal his identity: Jeff Morris (Reviewer #1).

The paper provides an interesting genomic analysis of the nitrogen assimilation pathways in multiple strains of *Prochlorococcus* and both reviewers and I recommend that the paper be accepted for publication in *eLife* following minor revisions. Reviewer 1 had some comments about the presence of NarB and absence of NirA in some strains. Clearly you and your co-authors have thought of this but clarification in the paper is required. More critically, the assimilation of nitrate comes at the expense of (ultimately) photosynthetically generated electrons required to reduce nitrate to ammonium. This is hardly a new idea (see, e.g., Falkowski and Stone, 1975; Myers, 1980).

A summary of the criticisms is as follows:

1) Subsection “Nitrate assimilation genes are found within a distinct subset of *Prochlorococcus* clades”, last paragraph: In this paragraph we are told that the downstream genes in nitrate assimilation – e.g. NirA – are found in all low-light adapted clades. While technically accurate this sentence is a bit misleading because it suggests that NirA is found in all genomes of the LL clades, but it's clearly spotty throughout the tree (Figure 1). We're told this is because of <100% genome recovery, hinting that the strains that appear to not have NirA really do have it, but it was missed during sequencing. This might very well be true but we don't know for sure, and language in this paragraph should be changed to make that clear. For instance "indicating" should be "suggesting".

2) Figure 1: Along the same lines as the above comment, the presence of many HL strains with NarB but not NirA is perplexing to me. What's the point of having NarB without NirA? Is this also an artifact of incomplete genome sequencing, or an intermediate stage en route to losing both? While this figure is already very data-rich, it might be worth trying to incorporate genome completion percentages for each genome, so that the viewer can see that genomes missing one or the other gene are incompletely sequenced. Otherwise, I think it would be sufficient for the authors to comment on the possibility that NarB+ NirA- observations are caused by incomplete coverage.

3) Figure 3: The difference between Figure 3 and "Figure 3—figure supplement 1" is not obvious. I gather that one is a phylogeny made from amino acid alignments and the other is from nucleotides? Are the same genes/proteins used in each? If so, what's the advantage of using one vs. the other? Why not just use the gene phylogeny, which has added phylogenetic signal (from redundant 3rd codon position bp's)?

4) Did the authors try to do the ancestral state reconstruction to determine if the ancestral *Prochlorococcus* clade did have the nitrate assimilation genes. The authors assumed that the ancestral *Prochlorococcus* lineage had the nitrate assimilation gene clusters, similar to *Synechococcus*. Although if this transition could be described in more detail, that would help the general understanding of the results and putting them into broader context.

5) The discussion of the selection pressures at different depths was not completely clear. For example, why would a diversification in the brightly lit but nutrient poor environments (subsection “Macroevolutionary implications of differential trait variability in *Prochlorococcus* clades and ecotypes”, first paragraph) lead to an increase in light harvesting? If light availability is not low, there would not necessarily be an increase in light harvesting efficiency. Instead, there should be a selection for a reduction in nutrient requirements or for a more efficient nutrient uptake.

6) It is not clear why the nitrate assimilation capability, which is costly, would be selected for in the low nitrate environment. In addition to energetic and iron costs, is there a N cost to maintain the nitrate assimilation machinery? I can see how nitrate assimilation ability could be selected for in the environment where nitrate is present and there is enough light to cover the energetic costs (at intermediate depths?). Therefore, the presence of the ecotypes capable of nitrate assimilation both close to the surface (in HL lineages) and at greater depths (in a LL lineage) (e.g., Figure 7) is puzzling, as those environments should have different selective pressures. A discussion of the nitrate assimilation selection at different depths could be helpful. What is known about the frequencies of such ecotypes at different depths?

---

## [Author Response]

The paper provides an interesting genomic analysis of the nitrogen assimilation pathways in multiple strains of Prochlorococcus and both reviewers and I recommend that the paper be accepted for publication in eLife following minor revisions. Reviewer 1 had some comments about the presence of NarB and absence of NirA in some strains. Clearly you and your co-authors have thought of this but clarification in the paper is required. More critically, the assimilation of nitrate comes at the expense of (ultimately) photosynthetically generated electrons required to reduce nitrate to ammonium. This is hardly a new idea (see, e.g., Falkowski and Stone, 1975; Myers, 1980).

We thank the editors and reviewers for their valuable and constructive criticism of our manuscript. We agree that our discussion of the costs related to nitrate assimilation was not extensive enough and simplifying these costs to reducing power equivalents alone is not sufficient. Also, in focusing on more recent literature related to *Prochlorococcus*, we neglected to reference the rich collection of literature describing the relationship between photosynthesis and nitrate assimilation as well as the associated impacts on growth and efficiency. We have corrected these omissions in our revised manuscript as follows:

“Most *Prochlorococcus*, however, lack this genetic repertoire (Moore et al., 2002; García-Fernández et al., 2004; Berube et al., 2015). […] In *Prochlorococcus*, these costs appear to be observed as a decrease in growth rate under saturating light intensity when supplied with nitrate compared to ammonium as the sole nitrogen source (Berube et al., 2015).”

A summary of the criticisms is as follows:1) Subsection “Nitrate assimilation genes are found within a distinct subset of Prochlorococcus clades”, last paragraph: In this paragraph we are told that the downstream genes in nitrate assimilation – e.g. NirA – are found in all low-light adapted clades. While technically accurate this sentence is a bit misleading because it suggests that NirA is found in all genomes of the LL clades, but it's clearly spotty throughout the tree (Figure 1). We're told this is because of <100% genome recovery, hinting that the strains that appear to not have NirA really do have it, but it was missed during sequencing. This might very well be true but we don't know for sure, and language in this paragraph should be changed to make that clear. For instance "indicating" should be "suggesting".

We agree that the presentation of the results in this paragraph were not clear and could indeed be misinterpreted. As recommended, we have changed “indicating” to “these data suggest”. In addition we clarify the distribution of *nirA* among low-light adapted *Prochlorococcus* in our revised manuscript:

“Among all isolate and single cell genome assemblies for low-light adapted *Prochlorococcus* in our data set (82% average genome recovery), 73% contained the genes for the downstream half of the nitrate assimilation pathway – encoding machinery for the transport and reduction of nitrite to ammonium (Figure 1). […] While *nirA* (nitrite reductase) was broadly distributed across low-light adapted *Prochlorococcus* genomes, only those belonging to the LLI clade were observed to also possess *narB* (Figure 1).”

“We further observed that all sequenced isolates belonging to the LLI clade possess *nirA* and the average genome recovery of isolate and single cell genome assemblies belonging to the LLI clade matched the proportion of these assemblies with an annotated *nirA* gene (81% average genome recovery; 81% containing *nirA*). These data suggest that the ability to assimilate the more reduced nitrite is a core trait for cells belonging to the LLI clade.”

2) Figure 1: Along the same lines as the above comment, the presence of many HL strains with NarB but not NirA is perplexing to me. What's the point of having NarB without NirA? Is this also an artifact of incomplete genome sequencing, or an intermediate stage en route to losing both? While this figure is already very data-rich, it might be worth trying to incorporate genome completion percentages for each genome, so that the viewer can see that genomes missing one or the other gene are incompletely sequenced. Otherwise, I think it would be sufficient for the authors to comment on the possibility that NarB+ NirA- observations are caused by incomplete coverage.

Indeed, observations of strains with *narB*, but not *nirA*, are most likely due to the inherent incompleteness of single cell genomics (median 79% genome recovery in our data set). Further, while we were able to devise a PCR based screening assay to detect *narB* in the amplified single cell DNA, we were not successful at doing the same for *nirA* due to a lack of suitable primer sites for the design of degenerate primers. Ultimately, this means that we were capable of detecting more *narB* genes than *nirA* genes in the dataset, even when the *narB* and *nirA* genes were absent in the assembly due to uneven amplification across the genome. As insightfully suggested, we have modified Figure 1 by adding an additional ring that presents the genome recovery data.

3) Figure 3: The difference between Figure 3 and "Figure 3—figure supplement 1" is not obvious. I gather that one is a phylogeny made from amino acid alignments and the other is from nucleotides? Are the same genes/proteins used in each? If so, what's the advantage of using one vs. the other? Why not just use the gene phylogeny, which has added phylogenetic signal (from redundant 3rd codon position bp's)?

It is correct that Figure 3 and Figure 3—figure supplement 1 are phylogenies made from amino acid alignments and nucleotide alignments, respectively. The same genes/proteins and taxa are used in each figure. We respectfully disagree that gene phylogenies necessarily have added phylogenetic signal. While it is true that the 3rd position of the codon provides added phylogenetic resolution for closely related sequences, amino acid alignments can be advantageous when inferring phylogenies for more distantly related sequences. Over longer periods of evolutionary time, there is a greater likelihood for multiple substitutions and/or reversions at a single nucleotide site, which would be unaccounted for in a nucleotide model. Further, nucleotide models are simpler as they are based on a 4x4 matrix while amino acid alignments are more complex with a 20x20 matrix. Given that *Prochlorococcus* and *Synechococcus* are closely related, it’s possible that codon alignments would be entirely sufficient. On the other hand, there is a significant GC skew between the *Synechococcus* (~60% GC) and the high-light adapted *Prochlorococcus* (~30% GC). Each phylogeny provides a different perspective of the data set. As one example, the protein tree shows that the NirA from a HLVI clade single cell is most closely related to the Type II NirA of the LLI clade; but, in the gene tree, the *nirA* from the same HLVI clade single cell is grouped with the other high-light adapted genomes, possibly as a result of codon bias pressure towards AT rich codons in high-light adapted *Prochlorococcus*. In the interest of transparency, we would prefer to keep both Figure 3 and Figure 3—figure supplement 1 in the manuscript.

We fully appreciate, however, that the inclusion of both phylogenies made the presentation of our results difficult to navigate. Thus, to aid the reader in identifying the differences between these phylogenies, we have added the text “Amino Acid Phylogenies” and “Nucleotide Phylogenies” to the respective figures.

4) Did the authors try to do the ancestral state reconstruction to determine if the ancestral Prochlorococcus clade did have the nitrate assimilation genes. The authors assumed that the ancestral Prochlorococcus lineage had the nitrate assimilation gene clusters, similar to Synechococcus. Although if this transition could be described in more detail, that would help the general understanding of the results and putting them into broader context.

In developing our hypotheses, we did not originally attempt to reconstruct the state of *Prochlorococcus’* ancestor to assess the likelihood that it possessed the trait for nitrate assimilation. Our primary concern was that we obviously cannot have data for extinct low-light adapted *Prochlorococcus* that may or may not have possessed this trait. Instead, our assumption was rooted in both the high frequency of this trait in *Synechococcus* as well as the shared genomic location of nitrate and nitrite assimilation genes in both *Synechococcus* and low-light adapted *Prochlorococcus*. In some sense, we suppose this might be regarded as a qualitative ancestral state reconstruction. We understand the reviewer’s concern about this and appreciate her/his insightful suggestion. We have modified the manuscript to include an analysis of trait evolution using parsimony and maximum likelihood. We think this has improved the manuscript and has additionally provided context that informs the plausibility of our original assumption. We conclude that if gene loss has proceeded at a faster rate than gene gain – supported by our detailed analysis of the data set – the ancestral *Prochlorococcus* node likely possessed the trait for nitrate assimilation. These data are now presented in Figure 7—figure supplement 1 which includes an unweighted parsimony analysis, a parsimony analysis with a cost matrix defined to make gains 10 times more costly than losses, and an asymmetric maximum likelihood analysis. The underlying data set is provided as a source data file suitable for analysis in Mesquite to allow the reader to explore different ancestral state reconstruction parameters.

5) The discussion of the selection pressures at different depths was not completely clear. For example, why would a diversification in the brightly lit but nutrient poor environments (subsection “Macroevolutionary implications of differential trait variability in Prochlorococcus clades and ecotypes”, first paragraph) lead to an increase in light harvesting? If light availability is not low, there would not necessarily be an increase in light harvesting efficiency. Instead, there should be a selection for a reduction in nutrient requirements or for a more efficient nutrient uptake.

Thank you for pointing out the need to clarify our arguments. We agree that over evolutionary time, in environments that are nutrient poor, there should be selection for a reduction in nutrient requirement. Indeed there is widespread evidence of such “streamlining” evolution in *Prochlorococcus*. For example, its genomes are small and have an elevated AT content [AT base pairs require 1 fewer nitrogen atom than GC basepairs] (Dufresne et al., 2005, Genome Biology), its proteins contain fewer amino acids with nitrogen-rich side chains (Grzymski and Dussaq, 2012, ISME J), and its membranes are composed of glyco- and sulfolipids rather than phospholipids (Van Mooy et al., 2006, PNAS).

However, nutrient limitation also imposes an energetic cost. The free energy cost of nutrient uptake transport (∆rG) is given by:∆rG=RT ln([⁡n]_I/[n]_E)+zF∆ψ,

Where *R* is the gas constant, *T* is the temperature, [n]_I and [n]_E are the concentrations of nutrient n in the cell and in the environment, respectively, Z is the unit charge of nutrient n, F is the Faraday constant and ∆ψ is the membrane potential. Thus, as nutrient levels in the environment decrease, ∆rG increases and uptake transport requires stronger energetic driving. In other words, the oligotrophic oceans should in theory provide two linked selective pressures on photosynthetic cells: selection for a reduction in nutrient requirements *and* selection for an increase in the harvesting of solar energy (Braakman et al., 2017).

Finally, evolutionary reconstructions suggest that in this case theory matches observation. That is, a wide range of evidence from the metabolic network, the pigments and stoichiometry of the photosystems, ecology and light physiology alongside evidence on the macromolecular and elemental composition of cells (as cited above) suggest that over the course of evolution *Prochlorococcus* has steadily increased its cellular electron-to-nutrient flux ratio (ve/vn) (Braakman et al., 2017). Since photosynthetic electron flux carries solar energy into metabolism, a steady increase in ve/vn is consistent with dual selection pressures acting to decrease the nutrient requirement of cells and to increase the energetic driving of nutrient uptake (Braakman et al., 2017). To clarify these points in the manuscript, we have added the following text:

“This model can be understood by considering the free energy cost of nutrient uptake transport (∆rG), which is given by ∆rG=RT ln([⁡n]_I/[n]_E)+zF∆Ψ, where *R* is the gas constant, *T* is the temperature, [n]_I and [n]_E are the concentrations of nutrient ninside and outside the cell, respectively, Z is the unit charge of nutrient n, F is the Faraday constant and ∆Ψ is the membrane potential. […] Evolutionary reconstructions based on a wide range of evidence from the metabolic network and photosynthetic machinery, ecology and light physiology in turn suggests that over the course of evolution *Prochlorococcus* cells have responded to this selective pressure by steadily increasing their electron-to-nutrient flux ratio (ve/vn), which enhances the transfer of solar energy into metabolism (Braakman et al., 2017).”

6) It is not clear why the nitrate assimilation capability, which is costly, would be selected for in the low nitrate environment. In addition to energetic and iron costs, is there a N cost to maintain the nitrate assimilation machinery? I can see how nitrate assimilation ability could be selected for in the environment where nitrate is present and there is enough light to cover the energetic costs (at intermediate depths?). Therefore, the presence of the ecotypes capable of nitrate assimilation both close to the surface (in HL lineages) and at greater depths (in a LL lineage) (e.g., Figure 7) is puzzling, as those environments should have different selective pressures. A discussion of the nitrate assimilation selection at different depths could be helpful. What is known about the frequencies of such ecotypes at different depths?

We thank the editors and reviewers for pointing out that the discussion of selection for the nitrate assimilation trait was not clear in the manuscript. While selection pressures were discussed at length in a cited study (Berube et al., 2016), we agree that additional discussion would greatly help the reader by providing additional context from empirical studies. We have now updated our revised manuscript to read:

“Among high-light adapted *Prochlorococcus*, the temporal and spatial distributions of cells containing *narB* in the wild suggests that selection favors cells that can assimilate nitrate under conditions where the supply of light is enhanced and the overall supply of nitrogen is low (Berube et al., 2016). […] At even greater depths within the euphotic zone, low irradiance likely offsets any fitness benefit associated with nitrate assimilation, even when nitrate is abundant.”